# Ceapins inhibit ATF6α signaling by selectively preventing transport of ATF6α to the Golgi apparatus during ER stress

Ciara M Gallagher[1,2]*, Peter Walter[1,2]*

[1]Howard Hughes Medical Institute, University of California, San Francisco, San Francisco, United States; [2]Department of Biochemistry and Biophysics, University of California, San Francisco, San Francisco, United States

**Abstract** The membrane-bound transcription factor ATF6α is activated by proteolysis during endoplasmic reticulum (ER) stress. ATF6α target genes encode foldases, chaperones, and lipid biosynthesis enzymes that increase protein-folding capacity in response to demand. The off-state of ATF6α is maintained by its spatial separation in the ER from Golgi-resident proteases that activate it. ER stress induces trafficking of ATF6α. We discovered Ceapins, a class of pyrazole amides, as selective inhibitors of ATF6α signaling that do not inhibit the Golgi proteases or other UPR branches. We show that Ceapins block ATF6α signaling by trapping it in ER-resident foci that are excluded from ER exit sites. Removing the requirement for trafficking by pharmacological elimination of the spatial separation of the ER and Golgi apparatus restored cleavage of ATF6α in the presence of Ceapins. Washout of Ceapins resensitized ATF6α to ER stress. These results suggest that trafficking of ATF6α is regulated by its oligomeric state.

**\*For correspondence:** ciara@walterlab.ucsf.edu (CMG); peter@walterlab.ucsf.edu (PW)

**Competing interests:** The authors declare that no competing interests exist.

## Introduction

Activating transcription factor six alpha (ATF6α) is a type-II transmembrane protein localized in the endoplasmic reticulum (ER) where, with its close homolog ATF6β, it functions as an ER stress sensor in one of the three principal branches of the unfolded protein response (UPR) (*Haze et al., 1999*; *Gardner et al., 2013*). ATF6α target genes are exclusively cytoprotective, functioning to increase the folding capacity of the ER and restore ER homeostasis (*Adachi et al., 2008*; *Wu et al., 2007*). Cells or animals lacking ATF6α show impaired survival upon ER stress (*Wu et al., 2007*; *Yamamoto et al., 2007*). When demand exceeds the folding capacity of the ER, ATF6α is transported from the ER to the Golgi apparatus, where sequential cleavage by two Golgi-resident proteases – site-1 and site-2 proteases (S1P and S2P), respectively - releases its N-terminal domain (ATF6α-N) from the membrane as a functional b-Zip transcription factor. ATF6α-N is then imported into the nucleus (nuclear translocation) where it activates transcription of its target genes (*Ye et al., 2000*). The mechanism that retains ATF6α in the ER and then releases it to allow transport to the Golgi apparatus is unknown. Deciphering the mechanism of ATF6α's regulated trafficking is essential to understanding how proteostasis is maintained in the ER.

The mechanism whereby ATF6α senses ER stress has remained a mystery since its discovery in 1998 (*Yoshida et al., 1998*). The lumenal domain of ATF6α (ATF6α-LD) dictates whether ATF6α localizes to the ER or to the Golgi apparatus (*Chen et al., 2002*; *Schindler and Schekman, 2009*). In fact, the soluble ATF6α-LD alone is sufficient to sense ER stress (*Sato et al., 2011*), and attaching the ATF6α-LD to the constitutively transported SNARE protein Sec22 is sufficient to retain Sec22 in the ER of unstressed cells, allowing its trafficking to the Golgi apparatus only upon ER stress (*Schindler and Schekman, 2009*). Thus ATF6α-LD is ATF6α's stress-sensor and regulates its

**eLife digest** Newly made proteins must be folded into specific three-dimensional shapes before they can perform their roles in cells. Many proteins are folded in a cell compartment called the endoplasmic reticulum. The cell closely monitors the quality of the work done by this compartment. If the endoplasmic reticulum has more proteins to fold than it can handle, unfolded or misfolded proteins accumulate and trigger a stress response called the unfolded protein response. This increases the capacity of the endoplasmic reticulum to fold proteins to match the demand. However, if the stress persists, then the unfolded protein response instructs the cell to die to protect the rest of the body.

A protein called ATF6α is one of three branches of the unfolded protein response. This protein is found in the endoplasmic reticulum where it is inactive. Endoplasmic stress causes ATF6α to move from the endoplasmic reticulum to another compartment called the Golgi apparatus. There, two enzymes cut ATF6α to release a fragment of the protein that then moves to the nucleus to increase the production of the machinery needed to fold proteins in the endoplasmic reticulum.

In a related study, Gallagher et al. identified a group of small molecules called Ceapins, which inhibit ATF6α activity. Here, Gallagher and Walter investigate how Ceapins act on ATF6α. The experiments show that Ceapin causes ATF6α molecules to form clusters that prevent the protein from moving to the Golgi apparatus by keeping it away from the machinery that moves proteins between these compartments. When the enzymes that cut ATF6α are sent to the endoplasmic reticulum, Ceapin treatment no longer prevents ATF6α activation, which shows that these small molecules specifically inhibit the stress-induced movement of ATF6α. When Ceapins are washed out of cells, the ATF6α clusters fall apart and ATF6α can now move to the Golgi.

These experiments show that ATF6α is actively held in the endoplasmic reticulum by a mechanism that is stabilized by Ceapins. Gallagher and Walter propose that the small clusters of ATF6α in unstressed cells act to keep this protein in the endoplasmic reticulum. However, when cells experience stress, the ATF6α clusters fall apart to allow the protein to move to the Golgi. The next steps following on from this work are to find out what these clusters are, how they are influenced by endoplasmic reticulum stress and exactly how the Ceapins stabilize these clusters.

trafficking. It is unclear what aspect of the folding environment ATF6α-LD is sensing, or if ATF6α senses misfolded proteins directly.

The molecular machinery required to move ATF6α from the ER to the Golgi apparatus, the COPII coat, is positioned on the cytosolic side of the ER membrane. Activation of ATF6α therefore requires transmitting the signal from the ER lumen to the cytosol (*Schindler and Schekman, 2009*; *Nadanaka et al., 2004*). It is unknown whether ATF6α interacts with the COPII coat directly or requires a transmembrane adaptor or traffics by bulk-flow after release from a transport-incompetent state. SREBP, the membrane bound transcription factor that responds to low cholesterol, has an elegant trafficking mechanism involving binding of a retention factor, INSIG, to a transport factor, SCAP (*Brown and Goldstein, 2009*). Indeed it is SCAP, not SREBP, that binds to and senses cholesterol and this binding regulates the interaction of SCAP both with its retention factor INSIG and with the COPII coat (*Sun et al., 2007*; *Motamed et al., 2011*). SREBP itself neither senses the signal nor interacts with the trafficking machinery. Similarly, a putative ACAP (ATF6α cleavage activating protein) and/or a putative INSIG-like ATF6α retention factor may remain to be discovered.

To gain insight into the mechanism whereby ATF6α senses stress we performed high-throughput cell-based screens to identify small molecule modulators of ATF6α signaling. In the accompanying paper, we describe the identification of Ceapins, a class of pyrazole amides that inhibit selectively the processing of ATF6α by S1P and S2P in response to ER stress but not the other UPR sensors, including – surprisingly – ATF6β, or SREBP. Using Ceapins to interrogate each step of ATF6α activation, we show here that Ceapins prevent selection of ATF6α into COPII vesicles by retaining it in foci in the ER membrane. Removing the requirement for trafficking by bringing together substrate and proteases restored cleavage in the presence of Ceapins. Ceapins induce

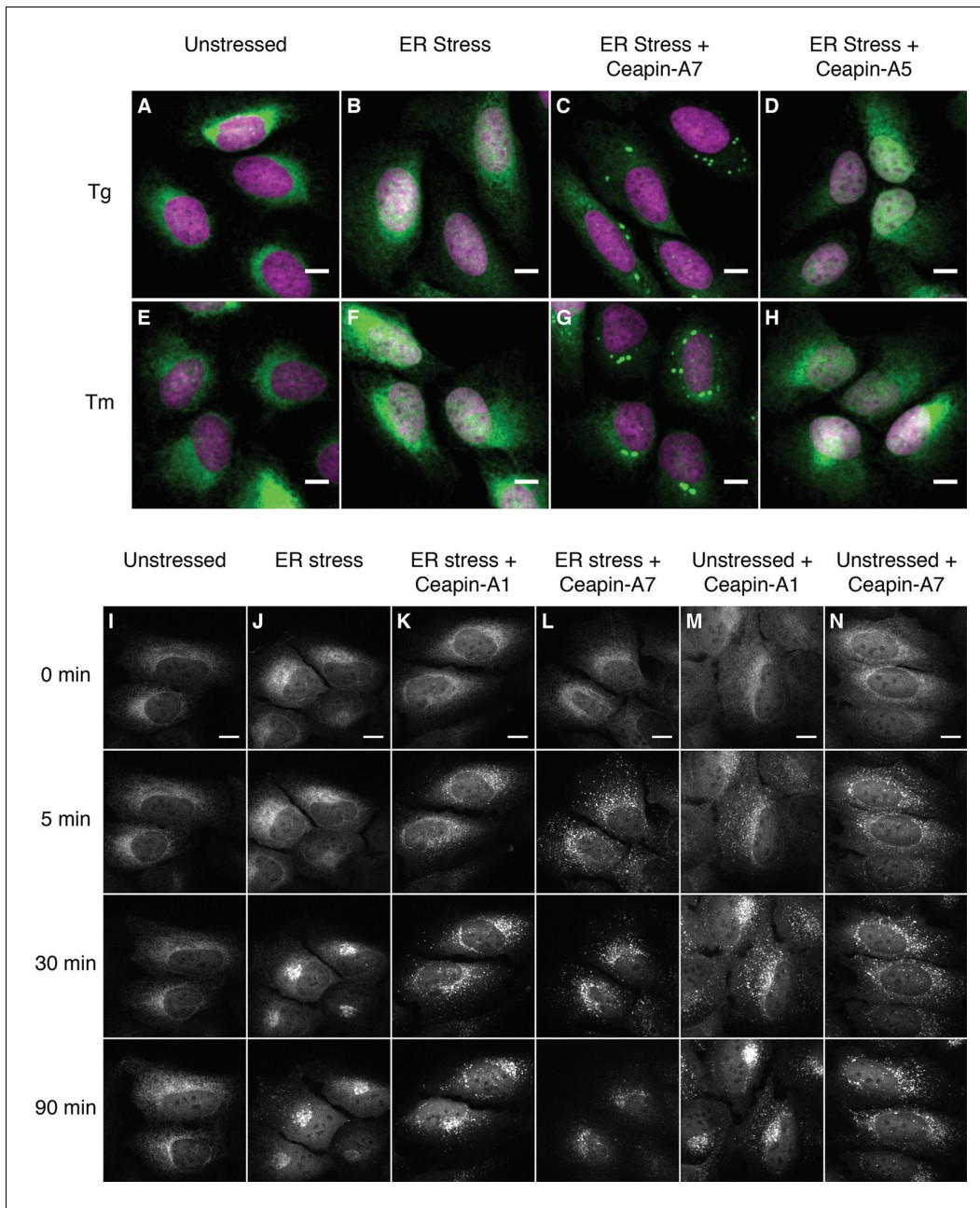

**Figure 1.** Ceapins induce foci formation and prevent ER-stress induced nuclear translocation of GFP-ATF6. (A–H) U2-OS cells stably expressing GFP-ATF6α were treated either with vehicle (A, E) or ER stress inducer (B–D and F–H) in the absence (A, B, E and F) or presence of active (6 µM Ceapin-A7, C, G) or inactive (6 µM Ceapin-A5, D, H) Ceapin analogs for five hours prior to fixation and fluorescent imaging of GFP-ATF6α (green) and DNA (magenta). In unstressed cells (A, E, DMSO) GFP-ATF6α is in the ER. Addition of either 100 nM thapsigargin (B) or 2.5 µg/mL tunicamycin (F) induces nuclear translocation of cleaved GFP-ATF6. The presence Ceapin-A7 (C, G) but not the inactive Ceapin analog A5 (D, H) prevents nuclear translocation. Scale bar is 10 µm. (I–N) Time-lapse images of U2-OS cells stably expressing GFP-ATF6α treated either with vehicle (I, DMSO), ER stress (J, 100 nM Tg), ER stress plus 5 µM Ceapin-A1 (K, IC50 4.9 ± 1.2 µM), ER stress plus 5 µM Ceapin-A7 (L, IC50 0.59 ± 0.17 µM) or Ceapin analogs alone (M, 5 µM Ceapin-A1; N, 5 µM Ceapin-A7). The addition of Ceapin analogs induces formation of GFP-ATF6α foci and either partially (K, Ceapin-A1) or completely (L, Ceapin-A7) inhibits nuclear translocation of GFP-ATF6α in response to ER stress. Scale bar is 10 µm.

The following figure supplements are available for figure 1:

*Figure 1 continued on next page*

*Figure 1 continued*

**Figure supplement 1.** Quantification of nuclear translocation assay with Ceapin Analogs.
**Figure supplement 2.** Active but not inactive analogs of Ceapin induce foci formation and prevent nuclear translocation of GFP-ATF6.
**Figure supplement 3.** GFP-ATF6 foci persist for up to 24 hr after addition of Ceapin A7.

rapid clustering of ATF6α, indicating that its oligomeric state plays a key role to regulate its trafficking and thereby activation in response to ER stress. Based on its mode of action corralling ATF6α into ER-restricted foci, we named Ceapins after the Irish verb 'ceap,' meaning 'to trap' .

## Results

### Ceapins clusters GFP-ATF6α into foci, preventing its nuclear accumulation

To understand how Ceapins inhibit ATF6α activation and processing in response to ER stress, we monitored ATF6α trafficking. To this end, we used a U2-OS cell line, which stably expresses a fluorescent GFP-ATF6α fusion protein at low levels. We followed nuclear translocation of GFP-ATF6α-N, the proteolytic fragment resulting from GFP- ATF6α cleavage in response to ER stress. In unstressed cells, GFP-ATF6α localized to the ER (*Figure 1A and E*, *green*). As expected, after induction of ER stress by treatment with thapsigargin (Tg), which inhibits the ER calcium pump, or tunicamycin (Tm), which inhibits N-linked glycosylation, we observed a fraction of the GFP fluorescence in the nucleus, apparent by co-localization with DAPI-stained nuclear DNA (*Figure 1B and F*, DNA in *purple*, co-localization indicated by *white color* in overlay). Induction of ER stress in the presence of active Ceapin-A7 (*Figure 1C and G*) but not of the inactive Ceapin analog A5 (*Figure 1D and H*) prevented nuclear translocation of GFP-ATF6α-N and led to an accumulation of GFP fluorescence in discrete foci (quantified in *Figure 1—figure supplement 1*). We have previously shown [accompanying manuscript; *Gallagher et al., 2016*] that under these conditions active Ceapin analogs block ATF6α proteolysis, indicating that the foci correspond to a pool of uncleaved GFP-ATF6α.

To characterize foci formation further, we next followed the cells in real time using live-cell imaging prior to and after induction of ER stress (*Figure 1I–N*; *Figure 1*, *Videos 1–6*). Treatment with vehicle alone showed ER localization that did not change over time (*Figure 1I*). In contrast, after induction of ER stress GFP fluorescence first accumulated in a perinuclear region, consistent with movement of GFP-ATF6α to the Golgi apparatus, and then accumulated in the nucleus, consistent with proteolytic processing and nuclear import of the resulting GFP-ATF6α-N (*Figure 1J*). Addition of either active Ceapin-A7 or Ceapin-A1 induced rapid foci formation of GFP-ATF6α, while inhibiting nuclear accumulation (*Figure 1K and L*). In contrast, the inactive Ceapin analog A5 failed to induce foci formation (*Figure 1—figure supplement 2*). Importantly, we observed that active but not inactive Ceapin analogs induce GFP-ATF6α foci even in the absence of ER stress (*Figure 1M and N*, *Figure 1—figure supplement 2*) and these foci persist for up to twenty-four hours (*Figure 1—figure supplement 3*). These results suggest that Ceapins inhibit ATF6α signaling by capturing it in foci. Interestingly we also see foci in cells subjected to ER stress alone at later time points corresponding to the time point at which attenuation of ATF6α signaling would initiate (*Figure 1J*, 90 min time point and *Video 2*) (*Haze et al., 2001*; *Rutkowski et al., 2006*).

### Ceapin-induced foci are reversible and correlate with inhibition of ATF6

To assess if Ceapin-induced GFP-ATF6α foci depict a terminal state of ATF6α destined for degradation, we performed washout experiments and followed GFP-ATF6α foci using live cell imaging (*Figure 2* and *Videos 7–9*). Cells treated with active Ceapin analogs (Ceapin-A1 and Ceapin-A7; *Figure 2B and C*) showed rapid formation of GFP-ATF6α foci. We allowed foci to form for 17 min,

then washed the cells, and added media without inhibitors. Washout of both Ceapin analogs led to rapid dissolution of GFP-ATF6α foci, indicating the foci formation was reversible (*Figure 2B and C*). Cells treated with vehicle alone showed no change in GFP-ATF6α localization throughout the washout experiment (*Figure 2A*). We observed the same washout kinetics in cells pretreated for three hours with cycloheximide to inhibit protein synthesis, a time point at which it is reasonable to expect any newly translated GFP-ATF6α had folded and matured (*Heim et al., 1994*; *1995*; *Cormack et al., 1996*; *Li et al., 1998*; *Sacchetti, 2001*; *Sacchetti et al., 2001*; *Zhang et al., 2006*; *Pédelacq et al., 2006*; *Ugrinov and Clark, 2010*) (*Figure 2—figure supplement 1* and *Videos 10–13*). This result indicates that the same molecules of GFP-ATF6α clustered into foci by Ceapins are redistributed in the ER upon washout.

### Videos 1–6

Time-lapse imaging of U2-OS cells stably expressing GFP-ATF6α treated either with vehicle (*Video 1*, DMSO) or ER stressor (100 nM Tg) in the absence (*Video 2*) or presence active Ceapin analogs (*Video 3*, 10 µM Ceapin-A1), (*Video 4*, 1 µM Ceapin-A7) or with active Ceapin analogs alone (*Video 5*, 10 µM Ceapin-A1), (*Video 6*, 1 µM Ceapin-A7).

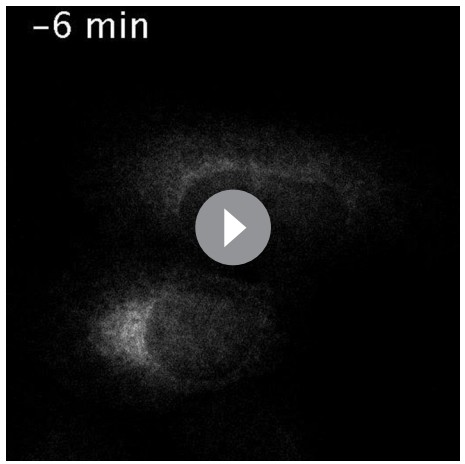

**Video 1.** GFP-ATF6α expressing U2-OS cells treated with vehicle.

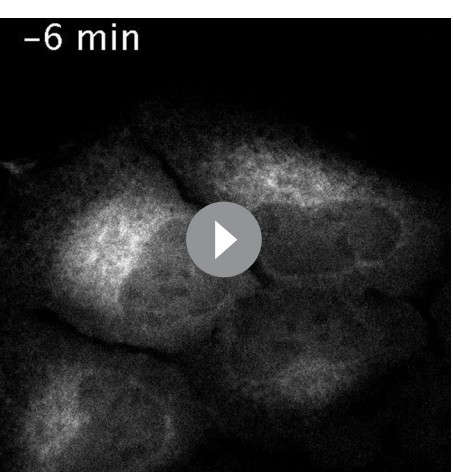

**Video 2.** GFP-ATF6α expressing U2-OS cells treated with ER stressor.

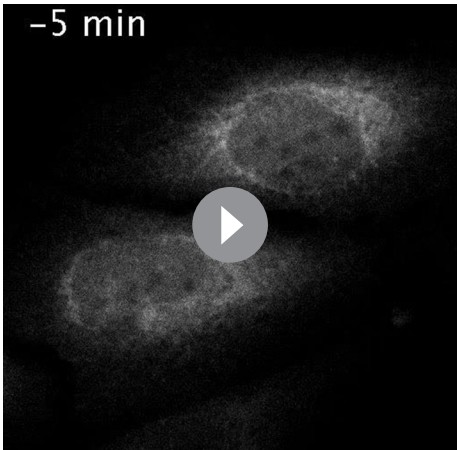

**Video 3.** GFP-ATF6α expressing U2-OS cells treated with ER stressor and Ceapin-A1.

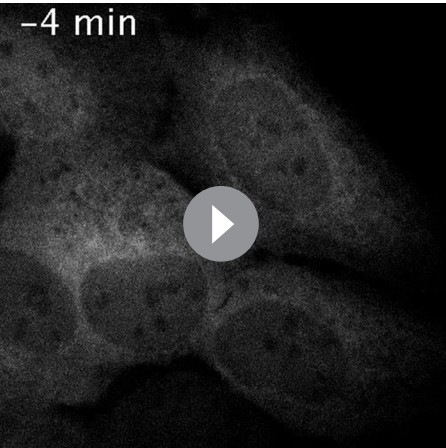

**Video 4.** GFP-ATF6α expressing U2-OS cells treated with ER stressor and Ceapin-A7.

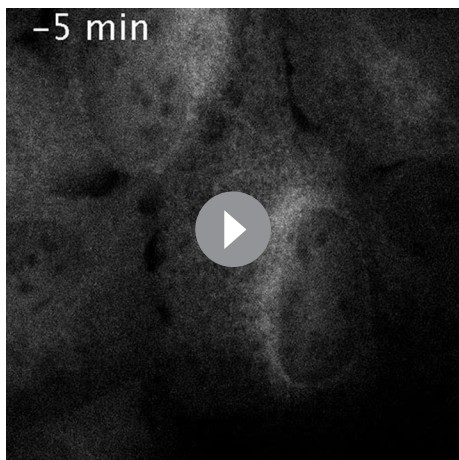

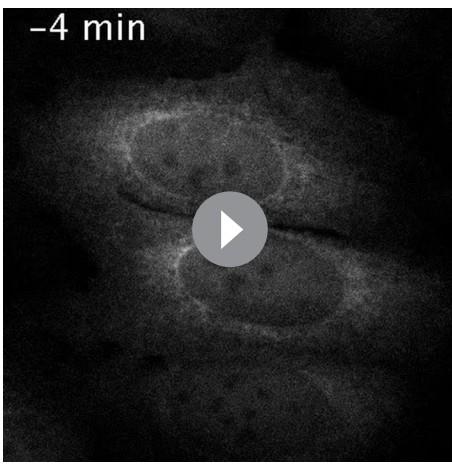

**Video 5.** GFP-ATF6α expressing U2-OS cells treated with Ceapin-A1.

**Video 6.** GFP-ATF6α expressing U2-OS cells treated with Ceapin-A7.

Images were acquired every minute and videos play at five frames per second. These videos are supplementary to *Figure 1*.

## Videos 7–9

Time-lapse imaging of U2-OS cells stably expressing GFP-ATF6α treated either with vehicle (*Video 7*, DMSO) or active Ceapin analogs (*Video 8*, 10 μM Ceapin-A1), (*Video 9*, 1 μM Ceapin-A7).

Seventeen minutes after compound addition cells were washed once with PBS and then fresh media without compound was added. Images were acquired every minute and videos play at five frames per second. These videos are supplementary to *Figure 2*.

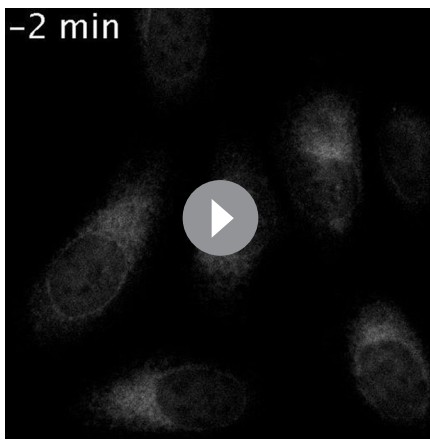

**Video 7.** Addition and washout of vehicle to GFP-ATF6α expressing cells.

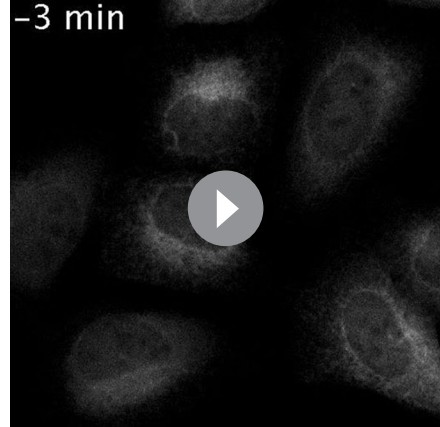

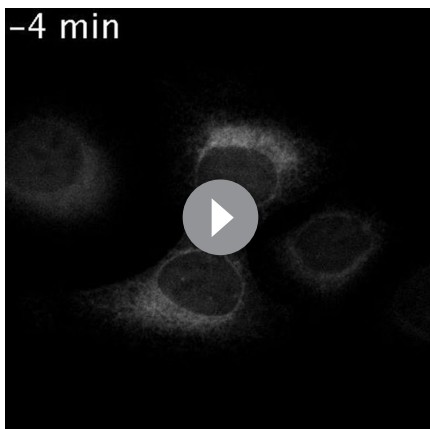

**Video 8.** Addition and washout of Ceapin-A1 to GFP-ATF6α expressing cells.

**Video 9.** Addition and washout of Ceapin-A7 to GFP-ATF6α expressing cells.

### Videos 10–13

Time-lapse imaging of U2-OS cells stably expressing GFP-ATF6α pretreated either with vehicle (*Video 10*, *Video 11*, Ethanol) or protein synthesis inhibitor (*Video 12*, *Video 13*, 0.1 µg/ml cyclo-heximide) for three hours prior to imaging.

During imaging, cells were treated either with vehicle (*Video 10*, *Video 12*, DMSO) or Ceapin-A1 (*Video 11*, *Video 13*, 10 µM Ceapin-A1). Sixteen minutes after compound addition cells were washed once with PBS ± protein synthesis inhibitor and then fresh media without compound ± protein synthesis inhibitor was added. Images were acquired every minute and videos play at five frames per second. These videos are supplementary to *Figure 2*.

To test if foci dissolution reflected the release of ATF6α from Ceapin-mediated inhibition, we repeated the washout experiment, except that after the washout the replacement media contained either vehicle or thapsigargin to induce ER stress (*Figure 2D–G*). After the washout, both control and Ceapin-treated cells responded to ER stress with the same kinetics, showing nuclear accumulation of GFP-ATF6α-N (*Figure 2E and G*, 85 min time point). Neither Ceapin-treated cells nor

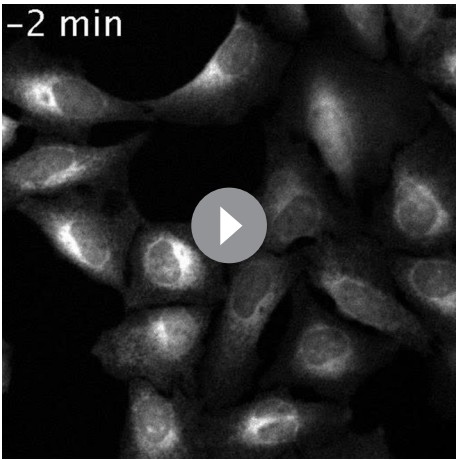

**Video 10.** Addition and washout of vehicle to GFP-ATF6α expressing cells pretreated with vehicle.

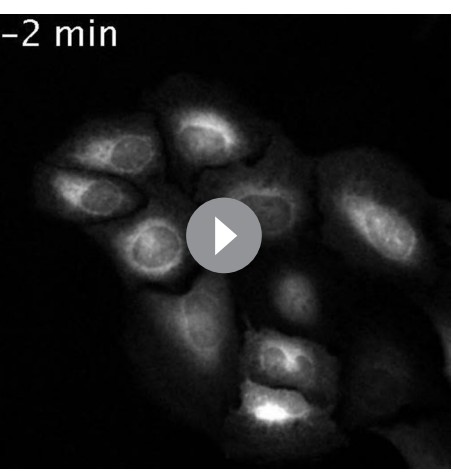

**Video 11.** Addition and washout of Ceapin-A1 to GFP-ATF6α expressing cells pretreated with vehicle

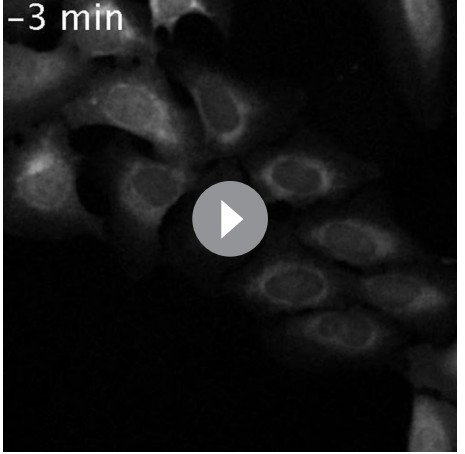

**Video 12.** Addition and washout of vehicle to GFP-ATF6α expressing cells pretreated with cycloheximide.

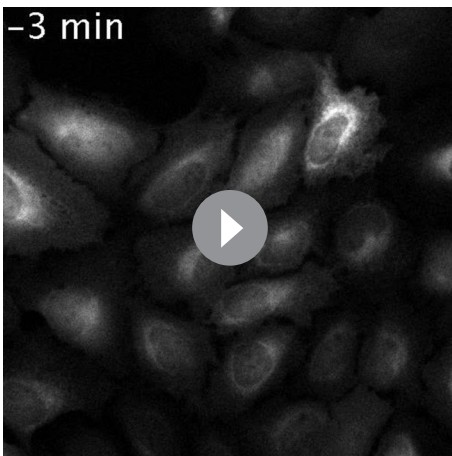

**Video 13.** Addition and washout of Ceapin-A1 to GFP-ATF6α expressing cells pretreated with cycloheximide.

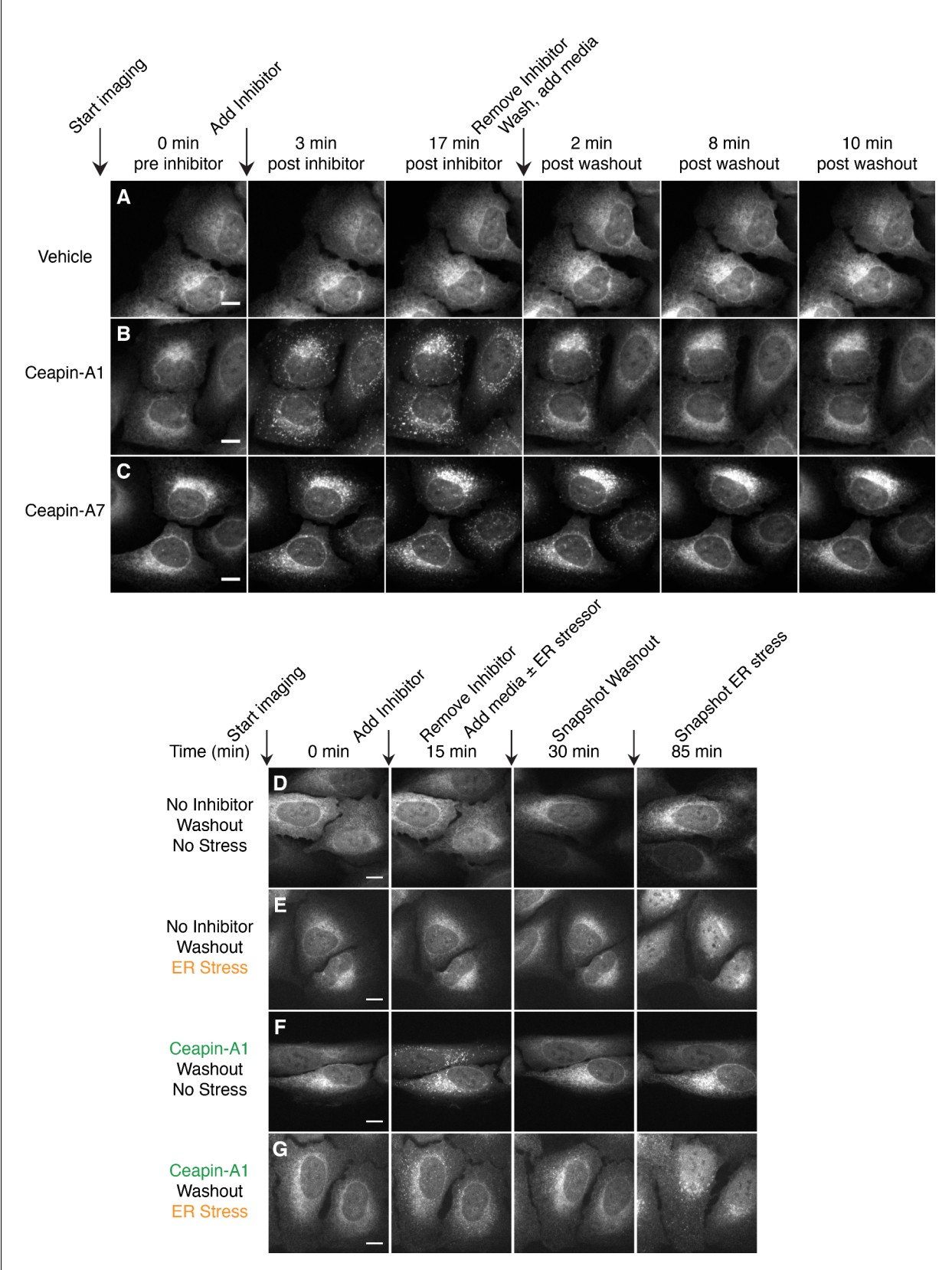

**Figure 2.** Ceapin-induced foci are reversible and correlate with inhibition of ATF6. (**A–C**) Time-lapse images of U2-OS cells stably expressing GFP-ATF6α treated either with vehicle (**A**, DMSO) or active Ceapin analogs (**B**, 10 μM Ceapin-A1), (**C**, 1 μM Ceapin-A7) for seventeen minutes to allow foci

*Figure 2 continued on next page*

*Figure 2 continued*

formation. Cells were then washed once with PBS and then media without compound was added. Scale bar is 10 µm. Images are representative of three independent experiments where three positions per well were imaged for each experiment. (D–G) Time-lapse images of U2-OS cells stably expressing GFP-ATF6α treated either with vehicle (A,B, DMSO) or 10 µM Ceapin-A1 (C,D). After fifteen minutes, the cells were washed with PBS and then media without (A,C, DMSO) or with ER stressor (B,D, 100 nM Tg) was added. After washout, ATF6α inhibitor foci resolve (C,D) and nuclear translocation of GFP-ATF6α occurs with similar kinetics in cells initially treated with either DMSO (B) or Ceapin-A1 (D). Scale bar is 10 µm. Images are representative of three independent experiments where three positions per well were imaged for each experiment.

The following figure supplement is available for figure 2:

**Figure supplement 1.** Redistribution of GFP-ATF6 from foci into the ER after washout of Ceapin-A1 does not require protein synthesis.

vehicle-treated cells that received media without thapsigargin showed nuclear accumulation of GFP-ATF6α-N (*Figure 2D and F*, 85 min time point), confirming that the washout and imaging procedures alone did not induce ER stress. These results suggest that the dissolution of GFP-ATF6α foci restores an activatable pool of GFP-ATF6α. Thus, Ceapins are reversible inhibitors of ATF6α.

## Ceapin-induced GFP-ATF6α foci are located along ER tubules and do not move to the Golgi apparatus

To further examine the subcellular localization of the Ceapin-induced GFP-ATF6α foci, we fixed and stained cells with an antibody to giantin to mark the Golgi apparatus. In the absence of ER stress, we observed little overlap between GFP-ATF6α and giantin (*Figure 3A*; giantin staining in *purple*, see arrowheads in zoomed view). In contrast, in the presence of ER stress, we observed clear co-localization of GFP-ATF6α and giantin (*Figure 3B*). Golgi apparatus localization was markedly enhanced upon treatment of ER stressed cells with S1P inhibitor, which blocks the Golgi-resident protease that initiates ATF6α processing (*Figure 3C*). This result was expected as under these conditions GFP-ATF6α traffics to the Golgi apparatus, where it accumulates because it cannot be cleaved (*Okada et al., 2003*). In contrast, GFP-ATF6α foci formed in the presence of Ceapin-A1 and ER stress did not co-localize with giantin (*Figure 3D*). This result suggests that ATF6α is not cleaved in the presence of active Ceapin analogs because it does not traffic to the Golgi apparatus upon induction of ER stress.

We next asked whether our GFP-ATF6α fusion protein faithfully reported on ATF6α biology without influence from the GFP tag. To this end, we used immunofluorescence in 293 T-REx cells that stably express a doxycycline-inducible 3xFLAG-6xHis-tagged ATF6α (3xFLAG-ATF6α). In unstressed cells, we found 3xFLAG-ATF6α in the ER, co-localizing with the ER marker calnexin (*Figure 3E*). Thirty minutes after induction of ER stress, a portion of 3xFLAG-ATF6α had moved out of the ER (*Figure 3F*, arrowheads in zoomed views). Treatment of these cells with Ceapin-A1, either in the presence (*Figure 3G*) or absence (*Figure 3H*) of ER stress, led to formation of 3xFLAG-ATF6α foci that decorated ER tubules (arrowheads in zoomed views). As expected, ER stress induced co-localization of 3xFLAG-ATF6α with giantin (*Figure 3J*). No such co-localization was observed upon co-treatment with Ceapin-A1 (*Figure 3K*). Thus the Ceapin-induced formation of ATF6α foci is independent of the nature of the tag and likely reflects an intrinsic property of ATF6α.

To ask to what degree Ceapins completely block or just delay GFP-ATF6α transport to the Golgi apparatus, we induced ER stress in U2-OS cells in the absence or presence of active Ceapin analogs and imaged cells after a prolonged 2.4 hr incubation. We stained cells for an ER (GRP94) and a Golgi apparatus (giantin) marker. In unstressed cells, GFP-ATF6α co-localizes with the ER marker with only marginal Golgi marker co-localization (*Figure 3M*). After 2.4 hr of ER stress, GFP-ATF6α showed pronounced Golgi apparatus localization with some ER staining remaining (*Figure 3N*). (Note that fixation conditions that best preserve ER structure are not optimal for nuclear staining, making nuclear translocation difficult to see in these images.) In the presence of active Ceapin analogs, GFP-ATF6α foci decorated ER tubules and show minimal co-localization with the Golgi marker (*Figure 3O and P*). Taken together, these results show that in different cell types and using different ATF6α variants, Ceapins induce ATF6α foci and prevent ATF6α trafficking to the Golgi apparatus in response to ER stress.

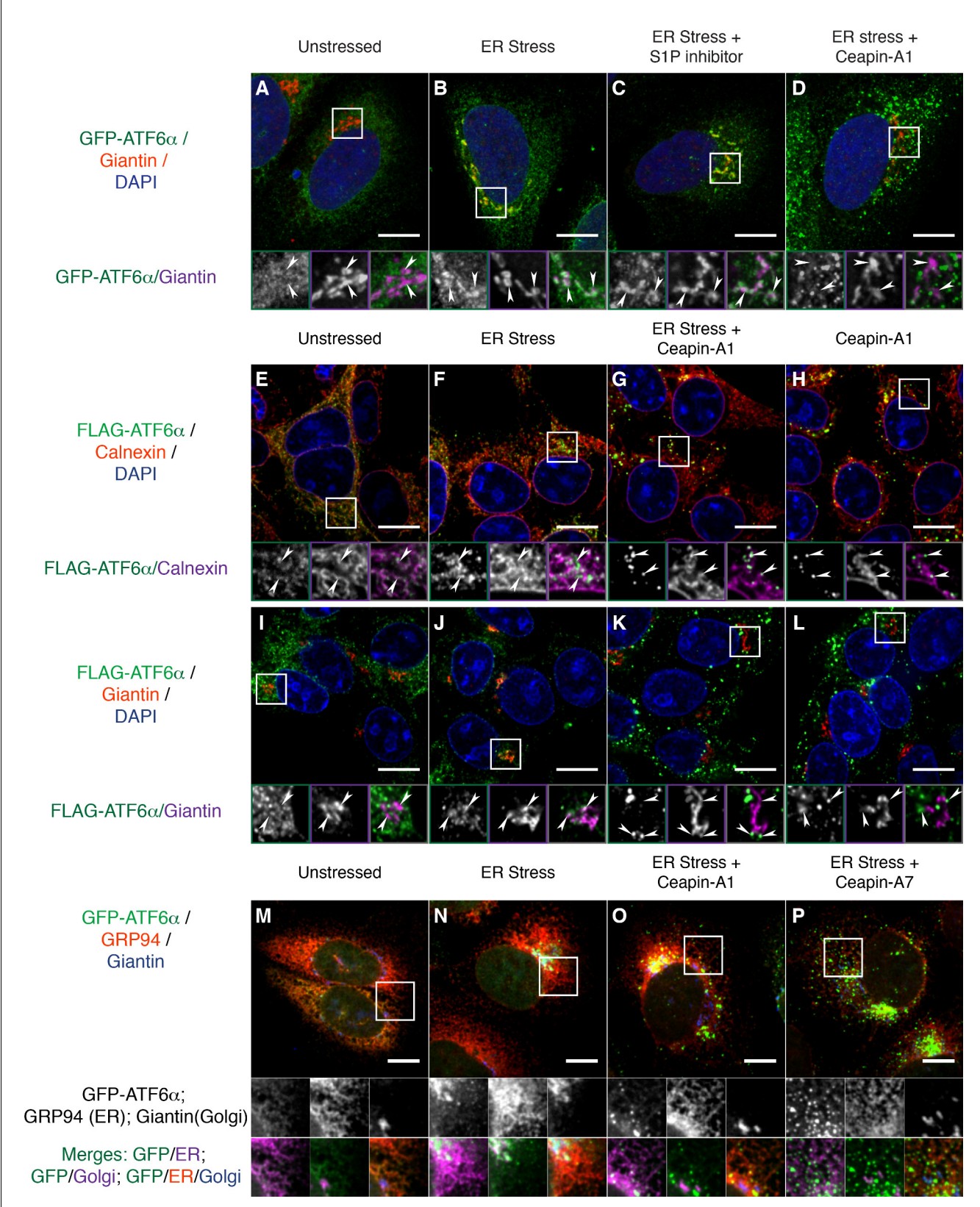

**Figure 3.** Ceapins retain tagged-ATF6α as foci in the ER and prevent trafficking of tagged ATF6α to the Golgi apparatus. (A--D) U2-OS cells stably expressing GFP-ATF6α were treated either with vehicle (A, DMSO), ER stress (B, 100 nM Tg), ER stress and site-1-protease inhibitor (C, 1 μM Pf-429242)
*Figure 3 continued on next page*

*Figure 3 continued*

or ER stress and Ceapin-A1 (**D**, 10 μM Ceapin-A1) for thirty minutes prior to fixation and fluorescent imaging of GFP-ATF6α (green), anti-Giantin to mark the Golgi apparatus (red in RGB, purple in GM insets) and DNA (blue). (**A**) Unstressed cells have minimal co-localization of GFP-ATF6α and Giantin. (**B**) ER stress induces trafficking of GFP-ATF6α to the Golgi apparatus where it colocalizes with Giantin. (**C**) ER stress combined with the site-1 protease inhibitor inhibits cleavage of GFP-ATF6α in the Golgi apparatus causing GFP-ATF6α to accumulate there. (**D**) ER stress combined with Ceapin-A1 shows minimal colocalization of GFP-ATF6α with Giantin, indicating GFP-ATF6α has not trafficked to the Golgi apparatus in the presence of the Ceapin-A1. (**E–L**) 293 T-REx cells stably expressing doxycycline inducible 3xFLAG-ATF6α were treated either with vehicle (**E,I**, DMSO), ER stress (**F,J**, 100 nM Tg), ER stress and Ceapin-A1 (**G,K**, 5 μM Ceapin-A1) or Ceapin-A1 alone (**H,L**, 5 μM Ceapin-A1) for thirty minutes prior to fixation and fluorescent imaging of 3xFLAG-ATF6α (green) and DNA (blue) and either an ER marker Calnexin (**E–H**, red in RGB, purple in GM inset) or a Golgi apparatus marker Giantin (**I–L**, red in RGB, purple in GM inset). ER stress induced Golgi trafficking of 3xFLAG-ATF6α (**J**, arrowheads) is prevented by the addition of the Ceapin-A1 (**K**). Ceapin-A1 either in combination with ER stress (**G**) or alone (**H**) induces formation of 3xFLAG-ATF6α foci that co-localize with ER tubules (arrowheads). (**M–P**) U2-OS cells stably expressing GFP-ATF6α were treated either with vehicle (**M**, DMSO), ER stress (**N**, 100 nM Tg), ER stress and active Ceapin analogs (**O**, 5 μM Ceapin-A1), (**P**, 5 μM Ceapin-A7). After time-lapse imaging for 2.4 hr, cells were fixed and stained for GFP-ATF6α (green), anti-GRP94 to mark the ER (red) and anti-Giantin to mark the Golgi apparatus (blue). ER stress induced trafficking to the Golgi apparatus (**N**) is blocked by the Ceapin analogs, and the induced GFP-ATF6α foci remain co-localized with ER tubules even after almost 2.5 hr of ER stress (**O,P**). Note that fixation conditions to visualize the ER and Golgi apparatus are not suitable for imaging the nuclear translocated fraction of GFP-ATF6α (see Materials and methods). Higher magnification panels underneath each image show each channel singly in greyscale (middle row), pairwise merges bottom row) of GFP-ATF6α (green) with either ER (magenta, bottom left) or Golgi markers in (magenta, and triple merge (bottom row, right) of GFP-ATF6α (green), ER (red) and Golgi (blue). In each panel, scale bars are 10 μm and boxed inserts are 7 x 7 μm (**A–L**) or 11.8 x 11.8 μm (**M–P**).

## Ceapins induce foci formation of endogenous ATF6

Using qPCR analysis, we showed in the accompanying manuscript that Ceapins inhibit ATF6α signaling. These analyses did not rely on the use of engineered reporters or over-expression of ATF6α [accompanying manuscript; *Gallagher et al., 2016*]. In contrast, the discovery of Ceapin-dependent ATF6α foci formation described here relied on the use of tagged and over-expressed fusion proteins. To address possible concerns arising from this approach, we next utilized a polyclonal antibody against ATF6α developed by the Mori lab (*Haze et al., 1999*) to image endogenous ATF6α in U2-OS cells (*Figure 4*). We already used this antibody to show that endogenous ATF6α is no longer proteolyzed in response to ER stress [accompanying manuscript; *Gallagher et al., 2016*].

We analyzed changes in ATF6α staining in response to ER stress in U2-OS cells. To our surprise, we found that, even in unstressed cells, endogenous ATF6α was not evenly distributed but found in small foci that were finely distributed over the ER network (*Figure 4A–D*). These observations contrasted with those described above made in cells over-expressing tagged versions of ATF6α that were diffusely ER-localized. After two hours of ER stress, a portion of endogenous ATF6α co-localized with the Golgi marker GM130 (*Figure 4B"*, arrowheads in inserts), and we observed a significant portion of ATF6α staining in the nucleus (*Figure 4B and B'*, quantified in *Figure 4E*, bar 2, $p < 0.0001$). Ceapin-A7 reduced co-localization of ATF6α with GM130 (GM130, *Figure 4C"*, arrowheads in inserts) and blocked nuclear accumulation of ATF6α, retaining it in ER foci (*Figure 4C'*, quantified in *Figure 4E*, bar 3, $p < 0.0001$). After two hours of treatment with Ceapin-A7 in the absence or presence of ER stress, ATF6α foci appeared larger and brighter than in unstressed cells (*Figure 4C,D*). The data show that endogenous ATF6α largely phenocopies the behavior of its tagged-variants characterized above and reveal that at physiological expression levels and in the absence of ER stress ATF6α is already clustered in small foci in the ER membrane. Ceapins may further stabilize these foci, trapping ATF6α in the ER and thus preventing its Golgi apparatus trafficking and nuclear translocation.

## Ceapins do not prevent ATF6α cleavage when ATF6α and proteases occupy the same organelle

Our results suggest a simple model: Ceapins inhibit ATF6α cleavage by antagonizing its transport to the Golgi apparatus, thereby preventing the encounter of substrate and proteases required to liberate ATF6α-N from the membrane. To test this notion, we treated 293 T-REx cells expressing 3xFLAG-ATF6α with brefeldin A (*Figure 5A*). Brefeldin A treatment fuses the Golgi apparatus and ER compartments (*Fujiwara et al., 1988*) and thus relocalizes both S1P and S2P proteases to the ER where they process ATF6α (*Shen and Prywes, 2004*).

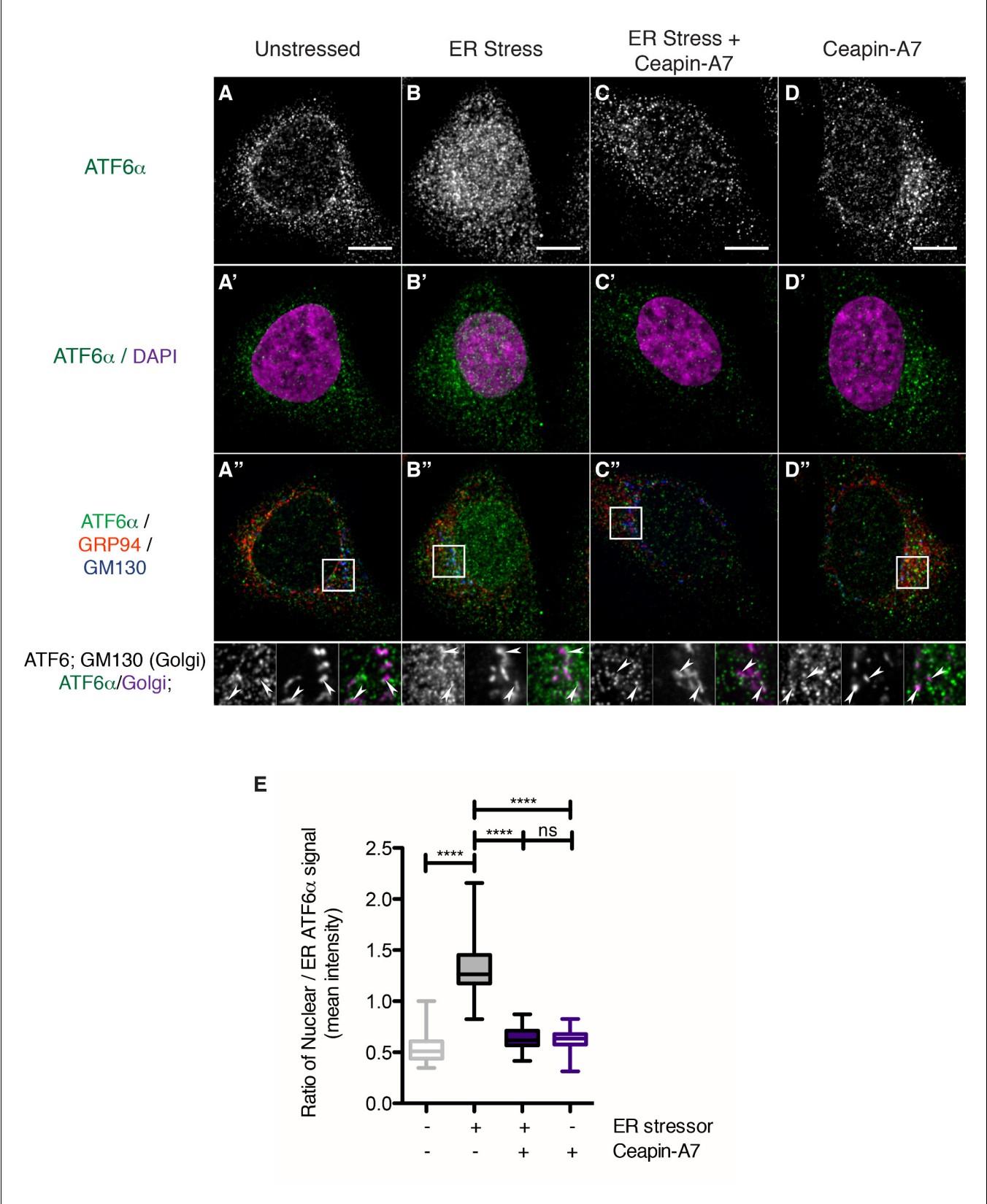

**Figure 4.** Endogenous ATF6α is in foci in unstressed cells and these foci are not changed by Ceapin-A7 either in the presence or absence of ER stress. (A–D) Nuclear translocation of endogenous ATF6α in response to ER stress is inhibited by Ceapin-A7. U2-OS cells treated with either vehicle (A, A',A")
*Figure 4 continued on next page*

*Figure 4 continued*

or ER stress (100 nM Tg), in the absence (**B,B',B"**) or presence of Ceapin-A7 (**C,C',C"**, 6 µM Ceapin-A7) or with Ceapin-A7 alone (**D,D',D"**) for two hours prior to fixation and fluorescent imaging of endogenous ATF6, anti-GM130 to mark the Golgi apparatus, GRP94 to mark the ER and DAPI to mark DNA. (**A–D**) Greyscale images of ATF6α for each treatment. (**A'–D'**) Merged images of ATF6α (green) and nuclear staining (purple). (**A"–D"**) Triple color merges of ATF6α (green) with ER (GRP94, red) and Golgi markers (GM130, blue). Boxed insets in (**A"–D"**) are shown below either as greyscale images of each channel (top row) or double (green/magenta) or triple (green/red/blue) merged images (bottom row). White arrows in boxed inserts point to Golgi staining. Scale bar is 10 µm and boxed inserts are 7 x 7 µm. (**E**) Quantification of nuclear translocation of endogenous ATF6. Plotted is the ratio of nuclear to ER intensity of ATF6α signal per cell as a box plot, whiskers are minimum and maximum values of the data. Statistics show the results of unpaired, two-tailed t-tests between indicated groups. Data plotted is from one of two independent experiments, each with at least twenty cells per treatment group.

As expected, ATF6α-N was produced upon treatment of cells with brefeldin A (*Figure 5A*, lanes 2, 12, 14, 24) or thapsigargin (*Figure 5B*, lanes 26, 36, 38, 48). Treatment with both S1P inhibitor and either brefeldin A (*Figure 5A*, lanes 7–10), or thapsigargin (*Figure 5B*, lanes 31–34), prevented ATF6α cleavage indicating that changing the subcellular localization of the protease had no effect on the efficacy of its inhibitor. In contrast, treatment of cells with active Ceapin analogs did not prevent production of ATF6α-N in brefeldin A-treated cells (*Figure 5A*, lanes 3–6 and lanes 15–18), whereas it did prevent production of ATF6α-N in thapsigargin-treated cells (*Figure 5B*, lanes 27–30 and lanes 39–42). As expected, the inactive Ceapin analog A5 had no effect on either treatment (*Figure 5A*, lanes 19–22 and *Figure 5B*, lanes 43–46). These results show that Ceapins do not convert ATF6α to an inaccessible or uncleavable form. Instead, cleavage of ATF6α in response to ER stress is inhibited because ATF6α does not traffic to the compartment where the proteases required for cleavage reside.

## Ceapins prevents selection of ATF6α into COPII vesicles

The foci of GFP-ATF6α observed with Ceapin treatment resemble the staining pattern observed for proteins that localize to ER exit sites. To test if in the presence of Ceapins GFP-ATF6α accumulated in ER exit sites, we co-stained for SEC16 and SEC31A, which are established markers of ER exit sites and the COPII coat assembling there (*D'Arcangelo et al., 2013*; *Hughes et al., 2009*) (*Figure 6A–E*), arrowheads in zoomed view mark ER exit sites that stained for both SEC16 and SEC31A). To this end, we induced ER stress in U2-OS cells expressing GFP-ATF6α in the absence or presence of Ceapin analogs. In unstressed cells, we observed minimal overlap of GFP-ATF6α foci and SEC16/SEC31A double-positive foci (*Figure 6A*, quantified in *Figure 6F*). When cells were treated with ER stress, the amount of overlap (indicated by triple-positive foci) increased (*Figure 6B and F*), consistent with GFP-ATF6α passing through ER exit sites on its way to the Golgi apparatus.

As expected, induction of ER stress in the presence of active Ceapin analogs produced GFP-ATF6α foci; however, these foci were non-overlapping with the SEC16/SEC31A positive ER exit sites (*Figure 6C and D*), while the inactive Ceapin analog A5 did not prevent co-localization of GFP-ATF6α with SEC16 / SEC31A (*Figure 6E*). In all cases, quantification confirmed the visual impression (*Figure 6F* and *Figure 6—figure supplement 1*). While we could observe a trend for fewer exit sites in ER stressed cells, the mean number of ERES counted per cell was not statistically different between conditions (*Figure 6G*).

Using live cell imaging, we followed GFP-ATF6α trafficking in response to ER stress in U2-OS cells that were transiently transfected with a fluorescent marker for ER exit sites, mRFP-p125A (*Klinkenberg et al., 2014*) (*Figure 6H*). At a thirty-minute time point after ER stress induction, GFP-ATF6α co-localized with mRFP-p125A (*Figure 6H*, top panels). In contrast, when we treated the cells with ER stressor and active Ceapin-A7, GFP-ATF6α foci remained distinct from mRFP-p125A foci (*Figure 6I*). Taken together with our previous data showing that cleavage of neither SREBP in response to low cholesterol nor ATF6β in response to ER stress, which also require COPII mediated transport from the ER to the Golgi apparatus and both S1P and S2P, are inhibited by Ceapins [accompanying manuscript; *Gallagher et al., 2016*], it is clear that Ceapins do not block COPII-mediated trafficking. Instead, our data suggest that Ceapins act to selectively prevent ATF6α being selected as cargo to leave the ER during ER stress (*Figure 7*).

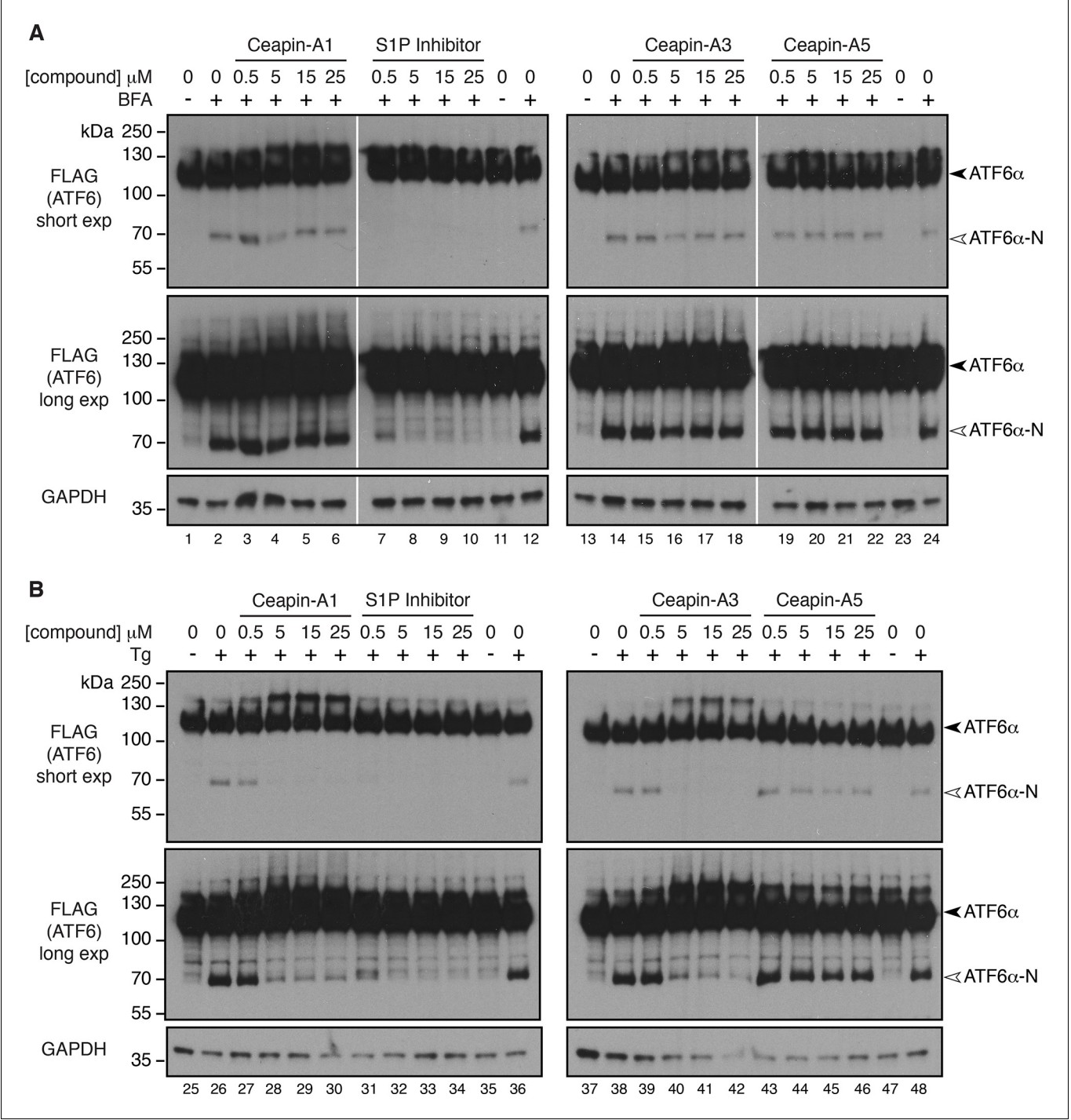

**Figure 5.** Collapsing the Golgi apparatus on to the ER restores cleavage of 3xFLAG-ATF6α in the presence of Ceapins. (**A**) 293 T-REx cells stably expressing doxycycline inducible 3xFLAG-ATF6α were treated either with vehicle (ethanol) or Brefeldin A (5 μg/mL BFA) in the absence or presence of increasing concentrations of either S1P inhibitor (Pf-429242) or active (Ceapin-A1, Ceapin-A3) or inactive (Ceapin-A5) Ceapin analogs for one hour prior to harvesting lysates for Western Blot analysis of 3xFLAG-ATF6α. (**B**) 293 T-REx cells stably expressing doxycycline inducible 3xFLAG-ATF6α were treated either with vehicle (DMSO) or ER stressor (100 nM Tg) in the absence or presence of increasing concentrations of either S1P inhibitor (Pf-429242) or active (Ceapin-A1, Ceapin-A3) or inactive (Ceapin-A5) Ceapin analogs for one hour prior to harvesting lysates for Western Blot analysis of 3xFLAG-ATF6α. For both A and B inhibitor concentrations were 0.5, 5, 15, 25 μM respectively. GAPDH is shown as a loading control. Black arrowheads – 3xFLAG-ATF6α, white arrowheads – 3xFLAG-ATF6α-N. White lines in A indicate where intervening lane has been removed.

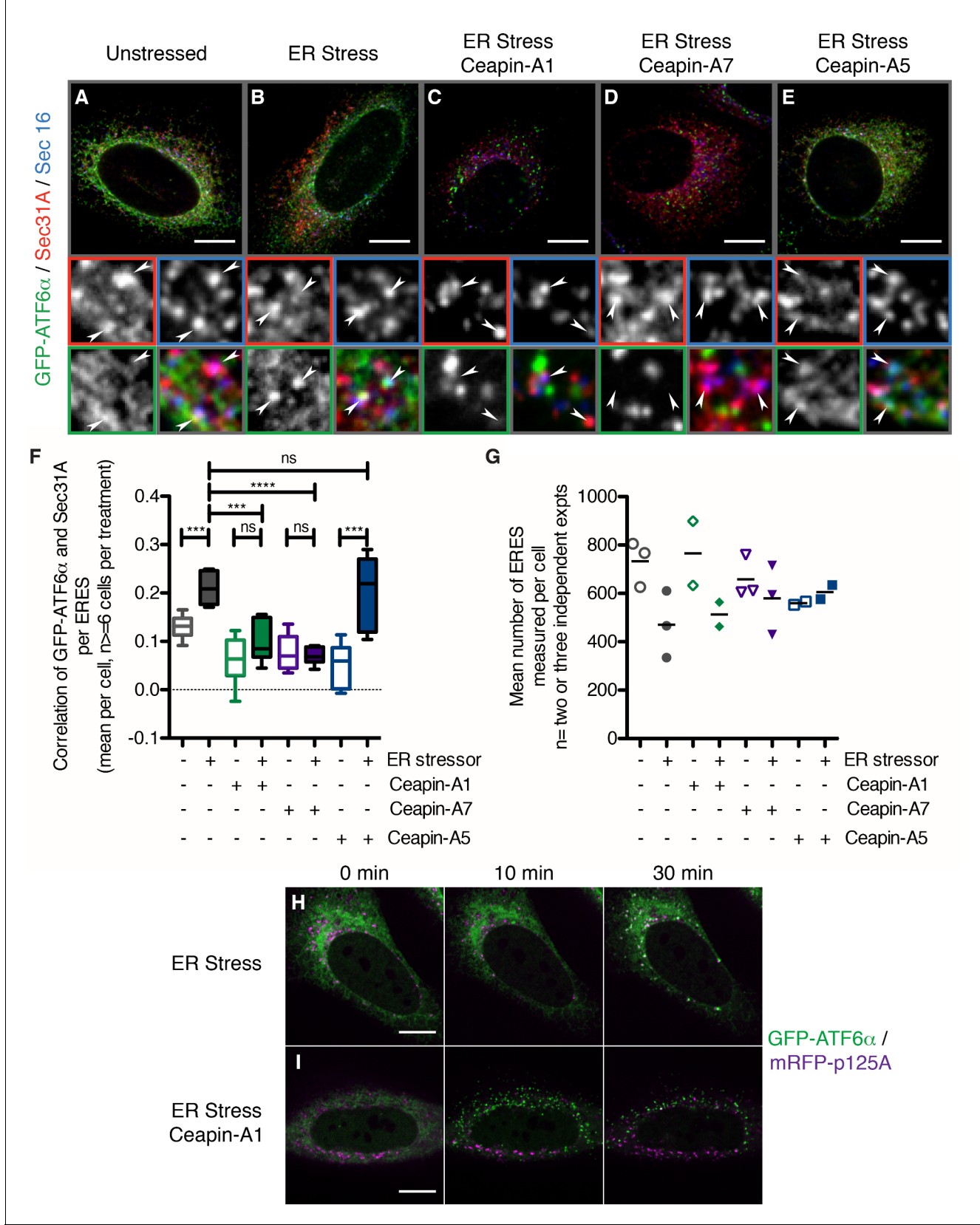

**Figure 6.** In the presence of Ceapins, GFP-ATF6α is no longer selected for transport from the ER in COP II vesicles. (A–E). U2-OS cells stably expressing GFP-ATF6α were treated either with vehicle (A, DMSO), or ER stressor (B, 100 nM Tg), in the absence or presence of either active (C, 10 μM

*Figure 6 continued on next page*

*Figure 6 continued*

Ceapin-A1, **D**, 1 µM Ceapin-A7) or inactive (**E**, 10 µM Ceapin-A5) Ceapin analogs for thirty minutes prior to fixation and fluorescent imaging of GFP-ATF6α (green), the COP II outer coat component Sec31A (red), the transmembrane ER exit site marker Sec16 (blue) and GRP94 to mark the ER (not shown). Punctae containing both Sec31A and Sec16 were denoted ER exit sites (ERES) and are marked with arrowheads in the inserts. Scale bar is 10 µm. Inserts are 3.3 µm x 3.3 µm or 39x zoom of lower magnification images. Note that the exposure times for GFP signal that is diffuse throughout the ER are not the same as those for GFP-ATF6α in foci (see Materials and methods). (**F**) Quantification of the correlation of GFP-ATF6α with Sec31A within Sec31A / Sec16 positive ERES for a single experiment where for at least 6 cells per condition, each cell imaged as a stack of 6 slices. Plotted is the mean and standard deviation of the mean per cell correlation of GFP-ATF6α and Sec31A. Statistical analysis used unpaired two-tailed t-tests. *** indicates p< 0.0009, **** indicates p= 0.0001, ns stands for non-significant. Variances were not different between treatments (F test). (**G**) The mean number of ERES analyzed per cells. Plotted is mean number of ERES measured per cell from either two or three independent experiments. None of the treatments were reproducibly statistically significantly different from each other. (**H-I**) Time lapse images of U2-OS cells stably expressing GFP-ATF6α (green) and transiently transfected with mRFP-p125A (purple) to mark ERES. Cells were treated with ER stressor (100 nM Tg) in the absence (**H**) or presence (**I**) of Ceapin-A7 and images were acquired at one frame every five minutes. Scale bar is 10 µm. Images are representative of at least three independent experiments.

The following figure supplement is available for figure 6:

**Figure supplement 1.** In the presence of Ceapins, GFP-ATF6 is no longer selected for transport from the ER in COP II vesicles.

## Discussion

The mechanism of ATF6α activation has remained poorly understood. Here we show that one of the earliest steps in ATF6α activation – its selection as cargo to leave the ER – can be inhibited using Ceapins, a newly identified chemical scaffold. In cells treated with Ceapins, ATF6α clusters in foci that do not leave the ER, because it no longer engages with ER exit sites upon ER stress. Consequently, ATF6α is trapped in its uncleaved, inactive state. Importantly, Ceapins no longer inhibit proteolysis and activation of ATF6α upon lifting the requirement for ATF6α trafficking from the ER for proteolytic processing by relocalizing the Golgi-resident S1P and S2P proteases back to the ER. This is in contrast to the S1P inhibitor, which as expected, inhibits S1P independently of its subcellular localization. Thus Ceapins do not drive ATF6α into an uncleavable form that would occlude the interaction with S1P.

This Ceapin-mediated inhibition of ER-to-Golgi trafficking is selective for ATF6α without inhibiting its closely related homolog ATF6β whose transport is similarly regulated by ER stress [accompanying manuscript; *Gallagher et al., 2016*]. This is the first evidence that there is a difference in the activation mechanisms employed by these two highly similar proteins. To date, ATF6α and ATF6β were thought to be equivalent in their mechanism of stress-sensing and trafficking. These data become even more surprising, because the lumenal domains of ATF6α and ATF6β are highly conserved and regulate ER stress-sensing and ER exit. Similarly, Ceapins do not interfere with SREBP signaling in response to cholesterol depletion, further underscoring that neither stress-sensing, cargo selection, nor ER-Golgi transport are generally affected by the drug. Ceapins could act on either side of the membrane to inhibit ATF6α trafficking. Its high level of selectivity strongly suggests that Ceapins act by binding to ATF6α or to an ATF6α–dedicated accessory factor that remains to be identified.

How ATF6α is retained in the ER in unstressed cells and how it is allowed to enter transport vesicles upon ER stress remains unknown. Our data showing that Ceapins trap ATF6α in the ER membrane in foci distinct from ER exit sites suggest a simple model of its action (*Figure 7*): Ceapins bind to ATF6α or some unknown accessory factor, stabilizing an oligomeric state that is not transport competent. In an oligomer, the interface required for interaction with the COPII coat could be buried, while in its monomeric form ATF6α would be recognized as cargo. This model is attractive because of the speed with which ATF6α foci form and dissolve after addition or removal of Ceapins (*Figure 2*). It is further supported by data showing that all active Ceapin analogs tested induced foci formation of ATF6α. In contrast, inactive Ceapin analogs either did not induce ATF6α foci or induced unstable foci that quickly disassembled during drug treatment. Moreover, dissolution of foci after washout of Ceapin analogs restored ER stress-induced transport to the Golgi apparatus.

Foci formation of ATF6α is also observed in the absence of Ceapin. We found that unlike overexpressed ATF6α fusion proteins, endogenous ATF6α in unstressed cells is found in foci, suggesting that the resting, ER-retained state of ATF6α is an oligomer or higher order complex. In addition,

ATF6α signaling attenuates in the face of unmitigated ER stress: although ATF6α is still plentiful in the ER, its proteolytic cleavage to produce ATF6α-N is only observed at early time points after induction of ER stress (*Haze et al., 2001*; *Rutkowski et al., 2006*). Consistent with this, we observed formation of ATF6α foci in cells treated with ER stress alone at time points at which attenuation of ATF6α signaling occurs. Our model poses that activation of ATF6α is achieved by shifting the equilibrium from a higher-order complex to a smaller entity (most likely an ATF6α monomer [*Nadanaka et al., 2007*]) that can be productively recruited into transport vesicles and that attenuation is achieved by shifting the equilibrium back. Ceapins prevent the formation of the transport-competent form even in the presence of ER stress. In this view, Ceapins engage the mechanisms that control normal retention/attenuation of ATF6α.

Our model is consistent with data showing that ATF6α exists in monomeric and oligomeric forms in the ER but that only the monomeric form is found in the Golgi apparatus (*Nadanaka et al., 2007*). Both stress-induced ER-Golgi trafficking and oligomerization of ATF6α are regulated by its stress-sensing lumenal domain. Thus the ATF6α ER-lumenal sensor domain would respond to stress in the ER by conversion from an oligomer to a monomer that would allow the information to be transmitted across the membrane to the cytosolic side initiating interactions with the COPII trafficking machinery. Among signaling transmembrane proteins there is precedence for conformational changes on one side of the bilayer leading to subsequent changes on the other side – notably for SCAP, the cholesterol sensor / trafficking adaptor of SREBP (*Sun et al., 2007*; *Motamed et al., 2011*), but also for the epidermal growth factor receptor (EGFR) (*Endres et al., 2013*). Conformational epitopes that regulate COPII mediated trafficking have also been shown for Sec22 (*Mancias and Goldberg, 2007*) and for the potassium channel Kir2.1 (*Ma et al., 2011*).

How ER stress is sensed by ATF6α is unknown. While it has been shown that ATF6α activation correlates with the release of BiP (*Shen et al., 2002*; *2005*), there is also evidence for ATF6α activation induced by direct ligand binding. These two principles are not mutually exclusive, as previously reconciled for the activation principles of IRE1 (*Pincus et al., 2010*). Evidence for activation by ligand binding comes from models of physiological ER stress in cardiac cells. In this system, ER stress induces expression of thrombospondin (Thbs4), which then binds to the lumenal domain of ATF6α and activates ATF6α signaling (*Lynch et al., 2012*). As with IRE1, ATF6α may bind newly accumulating misfolded proteins directly as ligands to activate signaling (*Gardner and Walter, 2011*). We envision that Ceapins act as antagonists to prevent such binding and stabilize ATF6α in its inactive state.

## Materials and methods

### Cell lines and culture conditions

Human bone osteosarcoma (U2-OS) cells were obtained from the American Type Culture Collection (ATCC HTB-96, ATCC Manassas, VA). U2-OS cells stably expressing GFP-HsATF6α were purchased from Thermo Scientific (084_01) and cultured with 500 µg/mL G418 (Roche 04 727 878 001) to maintain expression of GFP-ATF6α. 293 T-REx cells expressing doxycycline-inducible 6xHis-3xFLAG-HsATF6α are derived from (Tet)-ON 293 human embryonic kidney (HEK) cells (Clontech, Mountain View, CA) (*Cohen and Panning, 2007*) and are described elsewhere (*Sidrauski et al., 2013*) [accompanying manuscript; *Gallagher et al., 2016*]. Commercially available cell lines were authenticated by DNA fingerprint STR analysis by the suppliers. All cell lines were visually inspected using DAPI DNA staining and tested negative for mycoplasma contamination. Growth media was DMEM with high glucose (Sigma D5796) supplemented with 10% FBS (Life technologies, Carlsbad, CA # 10082147), 2 mM L-glutamine (Sigma G2150), 100 U penicillin 100 µg/mL streptomycin (Sigma P0781).

### Immunofluorescence of GFP-ATF6α in U2-OS cells (nuclear translocation assay)

Two days prior to compound addition 300 µL of $1.375 \times 10^4$ U2-OS-GFP-ATF6α cells per ml were plated per well in 96 well imaging plate (ibidi, Madison, WI 89626) and sealed with breathable seals (E&K Scientific, Santa Clara, CA T896100). Immediately prior to addition to cells, compounds were diluted to 6x in media from 500x DMSO stock and 60 µL 6x was added to cells for 1x final (0.2% DMSO).

After 5 hr, media was removed and cells were fixed in 4% paraformaldehyde (PFA) (Electron Microscopy Sciences, Hatfield, PA 15714) in PHEM buffer (60 mM PIPES, 25 mM HEPES, 10 mM EGTA, 2 mM MgCl2-hexahydrate, pH 6.9) for 15 min RT. Cells were permeabilized with PHEM-Tx (PHEM containing 0.1% Triton X-100, two washes, 5 min RT), washed twice in PHEM, blocked in PHEM containing 2% normal goat serum (Jackson Immunoresearch Laboratories, West Grove, PA, 005-000-121) for 1 hr RT. Primary antibodies were incubated in blocking solution overnight at 4 degrees. Cells were washed three times in PHEM-Tx then incubated with secondary antibodies and nuclear stain (DAPI, Molecular Probes, Eugene, OR, D-1306, 5 µg/mL) in blocking solution for 2 hr RT protected from light. Cells were washed three times PHEM-Tx, twice PHEM. Antibodies used were rat anti-GRP94 9G10 (abcam, Cambridge, MA ab2791), mouse anti-GFP 3E6 (Invitrogen, Carlsbad, CA, A11120), anti-rat-Alexa-555 (Invitrogen A21434), anti-mouse-Alexa-488 (Invitrogen A11029), each at 1:1000 dilution.

Plate was imaged on a spinning disk confocal with Yokogawa CSUX A1 scan head, Andor iXon EMCCD camera (Andor USA, South Windsor, CT) and 20x Plan Apo Objective NA 0.79 (Nikon, Melville, NY). Using the µManager high-content screening plugin 'HCS Site Generator' (*Edelstein et al., 2014*) 49 fields per well were acquired for mean cell number per well of 368 ± 12.

Images were analyzed using CellProfiler (*Carpenter et al., 2006*), MATLAB R2014a (Mathworks, Natick, MA) and GraphPad Prism version 5.0f (GraphPad Software, La Jolla, CA) as previously described [accompanying manuscript; *Gallagher et al., 2016*]. Masks for the ER and nucleus of each cell were created using the GRP94 and DAPI staining respectively. The ratio of the GFP intensity in the nucleus versus the ER was calculated for each cell and plotted as a histogram per well. A threshold for the minimum ratio of nuclear to ER signal corresponding to an activated (i.e. nuclear localized ATF6) cell was calculated as the minimum nuclear: ER ratio greater than 1 where the number of ER stressed cells (Tg) was greater than the corresponding unstressed control. Percent activation per well was calculated as the percentage of cells per well with a nuclear: ER ratio greater than the calculated threshold for that plate. Mean percent activation per well for a minimum of three replicate wells per treatment was plotted; error bars are 95% confidence limits. Compounds are annotated as hits if they show percent activation more than three standard deviations lower than the mean of ER-stress treated control.

## Live Imaging of GFP-ATF6α in U2-OS cells

U2-OS cells expressing GFP-ATF6α were plated at 7500 cells per well in eight-well ibiTreat µSlide (ibidi 80826) in growth media containing 500 µg/mL G418 two days prior to compound addition. For imaging, growth media was replaced with 250 µL imaging media per well. Imaging media is Leibovitz's L-15 medium, no phenol red (Life Technologies 21083–027) supplemented with 10% fetal bovine serum (heat-inactivated, Life Technologies 10082147), 2 mM L-glutamine (Sigma Aldrich G2150), 100 U penicillin and 0.1 mg/mL streptomycin (Sigma Aldrich P0781) and 0.45% glucose. Immediately prior to addition to cells, compounds were diluted to 6x in imaging media from 1000x DMSO stock and 50 µL 6x was added to cells for 1x final (0.2% DMSO). For longer time courses (>5 hr) final DMSO concentration was 0.034%.

Cells were imaged at 37 degrees Celsius on a spinning disk confocal with Yokogawa CSUX A1 scan head, Andor iXon EMCCD camera and 40x Plan Apo air Objective NA 0.95 with a 1.5x tube lens for additional magnification giving 60x final. Three positions per well were marked and imaged per experiment. Images were acquired using 488 nm laser at a rate of one frame per minute with 250 ms exposure time for each. At least five images were acquired per well prior to addition of compounds. Compounds were added between frames, two wells per addition. The frame preceding compound addition for each well was annotated t = 0 min.

## Washout of Ceapin analogs during live imaging of GFP-ATF6α in U2-OS cells

Cells were plated, vehicle or Ceapins were added and imaging was initiated as described above for live imaging. For washout, fifteen minutes after Ceapin analog or vehicle addition media was removed, wells were washed with 300 µL of PBS; PBS was removed and replaced with 300 µL of imaging media containing either DMSO or 100 nM Tg to induce ER stress. DMSO concentration was equal between all wells for both foci formation and induction of ER stress. Imaging used the 40x Plan Apo air Objective NA 0.95 either with or without the tube lens set to 1.5x additional magnification.

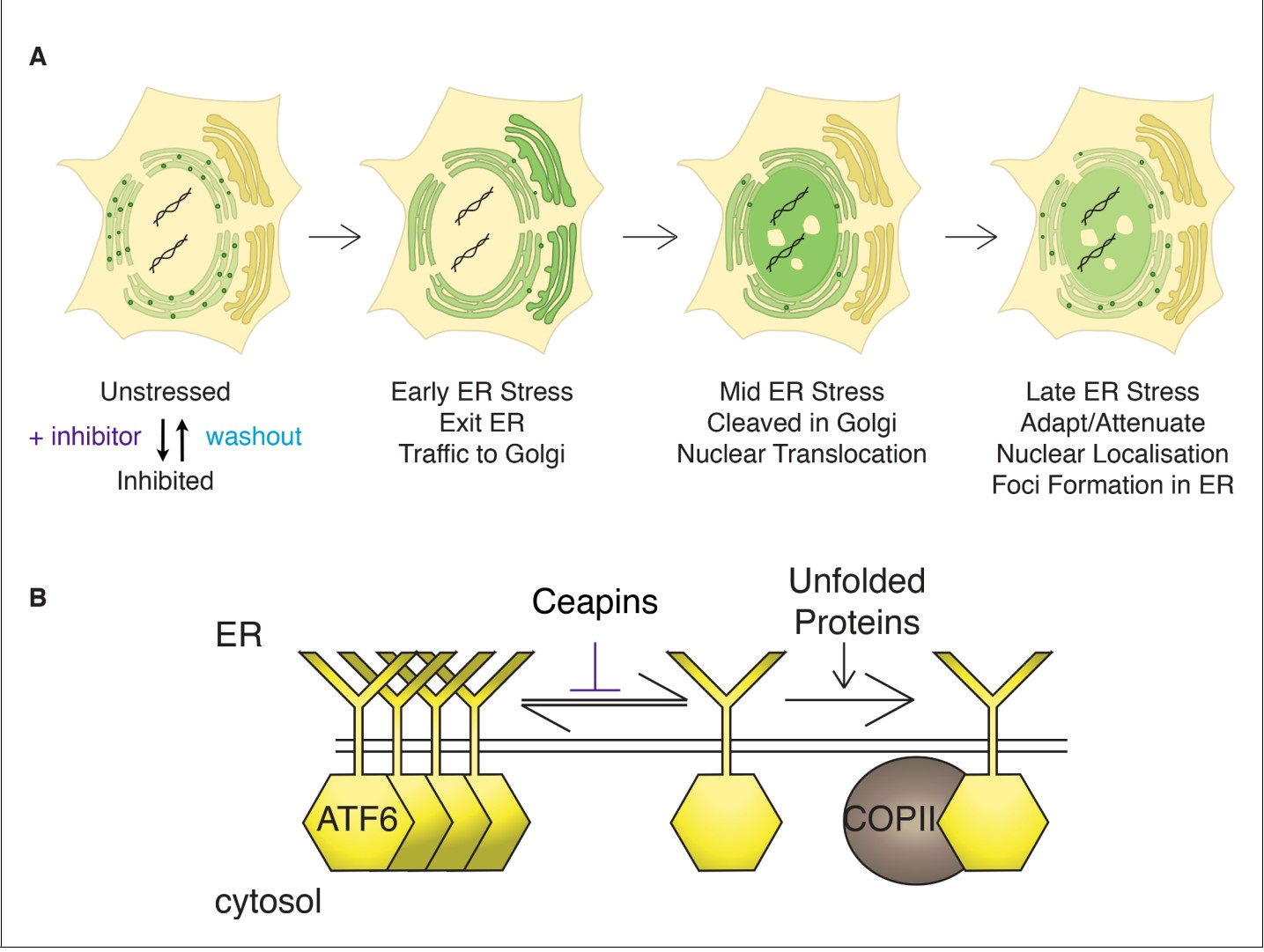

**A**

Unstressed | Early ER Stress Exit ER Traffic to Golgi | Mid ER Stress Cleaved in Golgi Nuclear Translocation | Late ER Stress Adapt/Attenuate Nuclear Localisation Foci Formation in ER

+ inhibitor ↕ washout
Inhibited

**B**

**Figure 7.** Model: Ceapins trigger an oligomeric state of ATF6α resembling the adapted / attenuated state. (**A**) Ceapin treatment inhibits COPII-mediated trafficking of ATF6a from the ER to the Golgi. In unstressed cells, ATF6α is in foci in the ER. Upon ER stress, ATF6a moves from the ER to the Golgi, where it is proteolyzed to release ATF6α-N that translocates to the nucleus and activates transcription of ATF6α target genes. After prolonged ER stress ATF6α signaling attenuates and ATF6α foci are seen in the ER. The appearance and stability of these foci are increased by Ceapin treatment in the absence or presence of ER stress. (**B**) In this model, Ceapin treatment stabilizes the inactive or attenuated form of ATF6a, which we hypothesize is an oligomeric state of ATF6α. In contrast, the monomeric form is stabilized by unfolded proteins and this form is capable of binding to the COPII coat and exiting the ER upon ER stress.

### Washout of Ceapin analogs during live imaging of GFP-ATF6α in U2-OS cells in the presence of protein synthesis inhibitors

Cells were plated in eight well μslides as described above for live imaging. Cells in growth media were treated with either 0.1 μg/ml cycloheximide in ethanol (Sigma C7698) or ethanol alone as vehicle for three hours prior to imaging. Prior to imaging, cells were washed once with 300 μL PBS containing either cycloheximide or ethanol and then placed in 250 μL imaging media containing either cycloheximide or ethanol. Immediately prior to addition to cells, compounds were diluted from 1000x DMSO stock to 6x in imaging media containing either cycloheximide or ethanol and 50 μL 6x was added to cells for 1x final. During imaging, cells were treated either with Ceapin-A1 (10 μM in DMSO) or DMSO alone as vehicle in the presence of either cycloheximide or ethanol. For washout, sixteen minutes after Ceapin analog or vehicle addition media was removed, wells were washed with 300 μL of PBS containing either cycloheximide or ethanol; PBS was removed and replaced with

300 µL of imaging media containing either cycloheximide or ethanol. Final ethanol and DMSO concentration was equal between all wells (0.1% each). Imaging used the 20x Plan Apo air Objective NA 0.75 and images were acquired once per minute. In the presence of ethanol, Ceapin-A7 induced foci did not wash out and so this analog could not be tested in this experiment.

## Immunofluorescence of 3xFLAG-ATF6α in 293 T-REx cells

293 T-REx cells expressing doxycycline-inducible 6xHis-3xFLAG-HsATF6a were plated at 20000 cells per well in an eight-well collagen IV coated µSlide (Ibidi 80822) in growth media one day prior to compound addition. After five hours, doxycycline (Sigma D9891) was added to 50 nM final. Immediately prior to addition to cells, compounds were diluted to 10x in media from 1000x DMSO stock and 30 µL 10x was added to cells for 1x final (0.2% DMSO).

After thirty minutes media containing compounds was removed and cells were fixed in 4% PFA in PHEM buffer as described above for U2-OS cells. Cells were permeabilized with PHEM containing 0.1% Triton X-100 (Sigma T9284) (PHEM-Tx), three washes each five minutes at room temperature. After two washes with PHEM cells were blocked in PHEM buffer containing 2% normal goat serum (Jackson Immunoresearch Laboratories 005-000-121) (blocking solution) for 1 hr RT. Primary antibodies were incubated in blocking solution overnight at 4 degrees. Cells were washed three times in PHEM-Tx then incubated with secondary antibodies in blocking solution for 2 hr RT protected from light. Cells were washed three times PHEM-Tx with the second wash containing 5 µg/mL DAPI. Prior to imaging cells were washed twice with PHEM buffer. Antibodies used were mouse anti-FLAG M2 (Sigma F1804), rabbit anti-Calnexin (Cell Signaling Technology, Danvers, MA, 2679S), rabbit anti-Giantin (abcam 24586), anti-mouse-Alexa-488 (Invitrogen A11029), anti-rabbit-Alexa-546 (Invitrogen A11010), each at 1:1000 dilution except anti-Calnexin, which was used at 1:100.

Slides were imaged on a spinning disk confocal with Yokogawa CSUX A1 scan head, Andor iXon EMCCD camera and 100x ApoTIRF objective NA 1.49 (Nikon).

## Immunofluorescence of GFP-ATF6α in U2-OS cells (visualization of ER / Golgi / ER exit sites)

U2-OS cells expressing GFP-ATF6α were plated at 7500 cells per well in eight-well ibiTreat µSlide (Ibidi 80826) in growth media containing 500 µg/mL G418 two days prior to compound addition. Immediately prior to addition to cells, compounds were diluted to 6x in media from 1000x DMSO stock and 50 µL 6x was added to cells for 1x final (0.2% DMSO).

After compound incubation (various times) media containing compounds was removed and cells were washed once quickly in PBS. Ice-cold methanol was added to fix and permeabilize cells and the slides were incubated for five minutes at minus thirty degrees Celsius. Cells were washed three times four minutes each with PHEM buffer and then blocked in PHEM buffer containing 2% normal goat serum (Jackson Immunoresearch Laboratories 005-000-121) (blocking solution) for 1 hr RT. Primary antibodies were incubated in blocking solution overnight at 4 degrees. Cells were washed three times in PHEM buffer then incubated with secondary antibodies in blocking solution for 2 hr RT protected from light. Cells were washed four times PHEM buffer, if necessary the second wash containing 5 µg/mL DAPI. Antibodies used were mouse anti-Sec31A (BD Biosciences, San Jose, CA, 612351), rabbit anti-Sec16 KIAA0310 (Bethyl Laboratories, Montgomery, TX, A300-648A), rabbit anti-Giantin (abcam 24586), rat anti-GRP94 9G10 (abcam ab2791), anti-mouse-Alexa-405 (Invitrogen A31553), anti-rabbit-Alexa-546 (Invitrogen A11010), anti-rabbit-Alexa-633 (Invitrogen A21071), anti-rat-Alexa-633 (Invitrogen A21094) each at 1:1000 dilution.

Slides were imaged on a spinning disk confocal with Yokogawa CSUX A1 scan head, Andor iXon EMCCD camera and 100x ApoTIRF objective NA 1.49 (Nikon). For analysis of ERES the exposure time for GFP-ATF6α in cells treated with active Ceapin analogs was shortened to prevent overexposure inflating the size of GFP-ATF6α foci. In unstressed and stressed controls, the GFP-ATF6α signal is distributed throughout the ER and is dimmer.

## Immunofluorescence of endogenous ATF6α in U2-OS cells

U2-OS cells (no reporters) were plated at 7500 cells per well in eight-well ibiTreat µSlide (Ibidi 80826) in growth media two days prior to compound addition. Immediately prior to addition to cells,

compounds were diluted to 6x in media from 1000x DMSO stock and 50 μL 6x was added to cells for 1x final (0.2% DMSO).

After two hours of compound treatment, cells were fixed and stained as for nuclear translocation assay in GFP-ATF6α expressing U2-OS cells (described above) with the following changes. Primary antibodies used were rabbit polyclonal anti-ATF6α (1:250, generous gift from Kazutoshi Mori), rat anti-GRP94 9G10, (1:1000, abcam ab2791) and purified mouse anti-GM130 clone 35 (1:250, BD Biosciences 610823. Secondary antibodies were all raised in goat and used at 1:1000 dilution - anti-rabbit-Alexa-488 (Invitrogen A11034), anti-rat-Alexa-555 (Life Technologies A21429) and anti-mouse-633 (Invitrogen A21050). Nuclear stain (DAPI, Molecular Probes D-1306, 5 μg/mL) was added in second of four PHEM-Tx washes after secondary antibody incubation and cells were washed twice in PHEM buffer before imaging in PHEM buffer. Slides were imaged on a spinning disk confocal with Yokogawa CSUX A1 scan head, Andor iXon EMCCD camera and 100x ApoTIRF objective NA 1.49 (Nikon).

Images were analyzed using CellProfiler (*Carpenter et al., 2006*), MATLAB R2014a and GraphPad Prism 5 as previously described [accompanying manuscript; *Gallagher et al., 2016*]. Masks for the ER and nucleus of each cell were created using the GRP94 and DAPI staining respectively. The ratio of the GFP intensity in the nucleus versus the ER was calculated for each cell and plotted as a boxplot. Statistics performed were unpaired two-tailed t-tests; similar results were obtained using one-way analysis of variance (ANOVA).

## Western blot analysis of 3xFLAG-ATF6α cleavage

Two days prior to drug treatment $2 \times 10^5$ 6xHis-3xFLAG-HsATF6α 293 T-REx cells per well were plated in 24 well plates (Corning, Corning, NY, 3526). The following day, expression of tagged ATF6α was induced using 50 nM doxycycline. Eighteen hours later either Brefeldin A (5 μg/mL final, in ethanol, Sigma Aldrich B6542) or ER stressor (100 nM Tg final, in DMSO, Sigma T9033) with or without inhibitors was added to cells and incubated for one hour. Vehicle was added to ensure the final concentration of either ethanol or DMSO was the same for all samples. Inhibitors used were S1P inhibitor (Pf-429242, Pfizer, New York, NY) or Ceapin analogs Ceapin-A1, Ceapin-A3 or the inactive Ceapin analog A5. All compounds were added from 1000x stock solutions. After one hour, media was removed, and 200 μL of scraping buffer (10 μM MG132 (Sigma C2211), 1x complete protease inhibitor (Roche Diagnostics, Pleasanton, CA, 05056489001) in phosphate buffered saline (PBS, Sigma Aldrich D8537)) was added to each well. Cells were scrapped into 1.5 mL eppendorf tubes, centrifuged at 3000 x g for five minutes at four degrees. Each cell pellet was resuspended in 50 μL 5x lysis buffer (200 mM Tris-HCl pH 8.0, 1% SDS, 40 mM dithiothreitol, 30% glycerol, pinch of bromophenol blue) supplemented with 10 μM MG132 and 1x complete protease inhibitor. Lysates were incubated on ice for twenty minutes, vortexed at full speed for five minutes at four degrees, incubated on ice for a further ten minutes, boiled for five minutes and centrifuged at 1000 x g for one minute at room temperature prior to loading. 7.5 μL of each sample was loaded on fifteen well ten percent mini-protean TGX gels (Bio-Rad Laboratories, Hercules, CA, 4561036). Gels were blotted onto 0.2 mM nitrocellulose membrane (Perkin Elmer, Santa Clara, CA, NBA083C00) and western blotted according to standard techniques. Blocking solution was 5% milk in PBS-Tween. Antibodies used were mouse anti-FLAG (M2, Sigma A2220) and rabbit anti-GAPDH (abcam ab9485), anti-mouse-HRP conjugate (Promega Corporation, Madison, WA, W4021) and anti-rabbit-HRP conjugate (Promega Corporation W4011). Horseradish peroxidase substrate (SuperSignal West Dura Extended Duration Substrate, Pierce Biotechnology, Rockford, IL, 34075) and Kodak X-OMAT film (Fisher Scientific, Waltham, MA, IB1651496) were used to detect protein bands.

## Quantification of GFP-ATF6α in ER exit sites (ERES)

Images were analyzed using CellProfiler 2.1.1. Briefly, the Sec31A and Sec16 images for each slice of each stack were multiplied so that only pixels containing both Sec31A and Sec16 fluorescence would be non-zero. In this resulting image, ERES were identified as objects with a diameter range of 0.167 – 1.67 μm. Thresholding was automatic and clumped objects were separated based on intensity. The resulting outlines of ERES were used as masks to count the intensity of Sec31A (405 nm), Sec16 (561 nm), GFP-ATF6α (488 nm) and GRP94 (633 nm) within ERES in the original images. The correlation between fluorophores was calculated on a pair-wise basis for all four and the results for correlation of Sec31A and GFP-ATF6α in double Sec31A / Sec16 positive punctae are shown in *Figure 6*. Data from

CellProfiler was imported into MatLab R2014a and organized by slice, cell and compound treatment. Results were imported into GraphPad Prism version 5.0 for statistical analysis and plotting. CellProfiler Pipelines and MatLab scripts are provided as source code files. Three independent experiments were analyzed.

## Live imaging of GFP-ATF6α and mRFP-p125A to visualize ER exit sites

One day prior to transfection, U2-OS cells stably expressing GFP-ATF6α were plated at a density of 7500 cells per well of an eight well ibiTreat μslide (Ibidi 80826) in growth media without 500 μg/mL G418. The following day, cells were washed once in PBS (Sigma Aldrich D8537) and growth media without antibiotics or selection agent was added. 2 ng of pmRFP-p125A (kind gift from Meir Aridor) and 78 ng of salmon sperm DNA (carrier DNA, Life Technologies 15632011) were transfected per well using Fugene HD transfection reagent (Promega E2311) in OptiMEM reduced serum media (Gibco 31985) according to manufacturers instructions. After six hours, media was replaced with growth media supplemented with 500 μg/mL G418 to ensure expression of GFP-ATF6. The transient transfection protocol was optimized to prevent transfection-induced ER stress (as visualized by ER versus nuclear localization of GFP-ATF6α in transfected cells) and to minimize over-expression of pRFP-125A to prevent distortion of ERES. Only unstressed cells with normal ER and ERES morphology were selected for imaging.

Twenty-four hours after transfection cells media was exchanged for imaging media (as described above). Cells were imaged on a spinning disk confocal with Yokogawa CSUX A1 scan head, a 100x Apo-TIRF objective NA 1.49 (Nikon) and a 565 nm long pass filter to split the emission light between two Andor iXon EMCCD cameras. For each cell, a z-stack with 0.25 μm steps was acquired every five minutes. Compounds were mixed to 6x from 1000x DMSO solutions in imaging media and added to cells for 1x final after acquisition of two z-stacks per cell. Post-acquisition, images from the cameras were aligned using the GridAligner plugin for ImageJ written by Nico Stuurman (http://valelab.ucsf.edu/~nstuurman/ijplugins/GridAligner.html) with the affine matrix calculated from reference images taken of a NanoGrid (an array of sub-wavelength sized holes, Miraloma Tech LLC, San Francisco, CA, A00011) using trans-illumination imaging. This imaging set-up is not suitable for long-term imaging of the cells.

## Acknowledgements

The authors thank Thomas Noriega for assistance with MatLab scripts used to analyze the imaging data, Christof Ozman for design and 3D printing of a microscope slide holder and Fran Sanchez for excellent technical assistance. We also thank Margaret Elvekrog and Voytek Okreglak for critical reading of the manuscript and the members of the Walter lab for discussion and support. The S1P inhibitor Pf-429242 was a generous gift from Pfizer. The plasmid for mRFP-p125A was a kind gift from Meir Aridor. HeLa-NF cells were a generous gift from Paul Wade (NIH).

## Additional information

### Funding

| Funder | Grant reference number | Author |
| --- | --- | --- |
| Howard Hughes Medical Institute | Investigator | Peter Walter |
| European Molecular Biology Organization | Long Term Fellowship | Ciara M Gallagher |

The funders had no role in study design, data collection and interpretation, or the decision to submit the work for publication.

### Author contributions

CMG, Conception and design, Data acquisition, Analysis and interpretation, Writing the manuscript; PW, Conception and design, Writing the manuscript

## Author ORCIDs

Peter Walter, http://orcid.org/0000-0002-6849-708X

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
