## [Decision Letter]

Thank you for submitting your work entitled "Ceapins inhibit ATF6α signaling by selectively preventing its transport to the Golgi apparatus during ER stress" for consideration by *eLife*. Your article has been evaluated by Randy Schekman as Senior Editor and three reviewers, one of whom, Davis Ng, is a member of our Board of Reviewing Editors.

The reviewers have discussed the reviews with one another and the Reviewing Editor has drafted this decision to help you prepare a revised submission.

The following individuals involved in review of your submission have agreed to reveal their identity: Jeffery Kelly (peer reviewer) and Peter Espenshade (peer reviewer).

Summary:

The manuscript from Gallagher and Walter focuses on the effects of the Ceapin class of inhibitors described in an accompanying manuscript. Here, the experiments are generally imaging approaches using GFP-tagged ATF6α (for time-lapsed live cell imaging) or by indirect immunofluorescence. First, the authors make clear that the direct molecular target of Ceapins remains unknown. Most of these data were obtained by GFP-ATF6α, which are convincing. Importantly, the authors analyze endogenous ATF6α using specific antibodies. Here, the data are somewhat different than recombinant overexpressed GFP-ATF6α. Instead of the diffuse distribution of GFP-ATF6α in resting cells, endogenous ATF6α displayed a wide ER distribution as small puncta. In the presence of Ceapin-A7, the distribution did not change much from control except that there was no shift to the Golgi/nucleus following ER stress. The authors also performed the classic Brefeldin A treatment to retrieve S1P and S2P to the ER where they processed ATF6α in the presence of Ceapins. The authors propose a model by which Ceapin molecules prevent the activation of ATF6α by causing it maintain clusters/aggregates in the ER and that prevent their partitioning into COPII vesicle budding sites upon ER stress.

1) Figure 1: There are fewer, larger puncta in these micrographs as well as significant amounts of signal diffuse in the ER. These appear different than other images with A7. Are the cells recovering after a 5 hr treatment?

2) Results, second paragraph: Addition of either active Ceapin-A7 or Ceapin-A1 induced rapid foci formation of GFP-ATF6α, while inhibiting Golgi apparatus localization and nuclear accumulation (Figure 1). Inhibition of Golgi localization cannot be concluded from this figure as no Golgi markers are shown. Ceapin-treated cells show puncta in a perinuclear location, not inconsistent with the Golgi apparatus.

3) Figure 2: In the washout experiments, cycloheximide should be added with the Ceapins to determine if the drug is reversible on the same molecules or alternatively, puncta are degraded and unclustered proteins after the washout represent the newly synthesized population.

4) Figure 5: Hand drawn MW marker marks on autoradiograms were not removed in some panels

5) ATF6α has two functional domains separated by the ER bilayer. Knowing which of these domains (or possibly both) is required for Ceapin inhibition would increase the impact of this study. Given that Ceapins do not inhibit activation of the homologous ATF6β, parallel studies using GFP-ATF6β and GFP-ATF6α/β chimeras would likely reveal which domain mediates Ceapin action, although still not testing whether this action is direct (This is not an essential experiment for full acceptance, but a suggestion).

6) Third paragraph, subsection “Ceapin-induced GFP-ATF6α foci are located along ER tubules and do not move to the Golgi apparatus”, figures numbers should be 3O and 3P.

7) Figure 7 just isn't intuitive, or useful to drive home the big picture, even after reading both of the linked manuscripts-maybe my printer didn't indicate ATF6α?

8) Shouldn't "its" in the title become "ATF6α"?

---

## [Author Response]

The manuscript from Gallagher and Walter focuses on the effects of the Ceapin class of inhibitors described in an accompanying manuscript. Here, the experiments are generally imaging approaches using GFP-tagged ATF6α (for time-lapsed live cell imaging) or by indirect immunofluorescence. First, the authors make clear that the direct molecular target of Ceapins remains unknown. Most of these data were obtained by GFP-ATF6α, which are convincing. Importantly, the authors analyze endogenous ATF6α using specific antibodies. Here, the data are somewhat different than recombinant overexpressed GFP-ATF6α. Instead of the diffuse distribution of GFP-ATF6α in resting cells, endogenous ATF6α displayed a wide ER distribution as small puncta. In the presence of Ceapin-A7, the distribution did not change much from control except that there was no shift to the Golgi/nucleus following ER stress. The authors also performed the classic Brefeldin A treatment to retrieve S1P and S2P to the ER where they processed ATF6α in the presence of Ceapins. The authors propose a model by which Ceapin molecules prevent the activation of ATF6α by causing it maintain clusters/aggregates in the ER and that prevent their partitioning into COPII vesicle budding sites upon ER stress.

1) Figure 1: There are fewer, larger puncta in these micrographs as well as significant amounts of signal diffuse in the ER. These appear different than other images with A7. Are the cells recovering after a 5 hr treatment?

We followed cells after addition of Ceapin-A7 for 24 hr and saw that foci coalesce and persist over time. We have now included these data as Figure 1—figure supplement 3. We also saw no evidence of recovery of ATF6 function in the accompanying manuscript where we show that cells treated with ER stressor in combination with Ceapin-A7 have higher rates of apoptosis compared to cells treated with ER stressor alone.

2) Results, second paragraph: Addition of either active Ceapin-A7 or Ceapin-A1 induced rapid foci formation of GFP-ATF6α, while inhibiting Golgi apparatus localization and nuclear accumulation (Figure 1). Inhibition of Golgi localization cannot be concluded from this figure as no Golgi markers are shown. Ceapin-treated cells show puncta in a perinuclear location, not inconsistent with the Golgi apparatus.

We removed the text “inhibiting Golgi apparatus localization and”. Golgi localization is better detailed in Figure 3.

3) Figure 2: In the washout experiments, cycloheximide should be added with the Ceapins to determine if the drug is reversible on the same molecules or alternatively, puncta are degraded and unclustered proteins after the washout represent the newly synthesized population.

We have added new data to address this question. We pretreated cells for three hours with cycloheximide, after which time any newly translated GFP-ATF6α has folded and matured. We then performed a washout experiment. We saw the same redistribution of GFP signal from punctae into the ER as in the absence of cycloheximide, indicating that Ceapins are reversible on the same molecules. These data have been included as Figure 2—figure supplement 2 and supplementary videos 10-13.

4) Figure 5: Hand drawn MW marker marks on autoradiograms were not removed in some panels

We have removed the markers and added a white line to indicate where the lane was cropped out.

5) ATF6α has two functional domains separated by the ER bilayer. Knowing which of these domains (or possibly both) is required for Ceapin inhibition would increase the impact of this study. Given that Ceapins do not inhibit activation of the homologous ATF6β, parallel studies using GFP-ATF6β and GFP-ATF6α/β chimeras would likely reveal which domain mediates Ceapin action, although still not testing whether this action is direct (This is not an essential experiment for full acceptance, but a suggestion).

We agree. We generated these chimeric constructs. Unfortunately over-expression of GFP-ATF6α through transient transfections does not reliably reproduce the biology of the endogenous protein – very few transfected cells show nuclear translocation in response to ER stressor and Ceapin induced foci are difficult to see given the high level of expression in the ER. It is difficult to infer Ceapin mediated inhibition or lack thereof for either ATF6α or the chimeras given the low level of nuclear translocation in the controls. This experiment will require substantial additional work and validation, which if successful will be included in a follow-up paper.

6) Third paragraph, subsection “Ceapin-induced GFP-ATF6α foci are located along ER tubules and do not move to the Golgi apparatus”, figures numbers should be 3O and 3P.

Thanks; done.

7) Figure 7 just isn't intuitive, or useful to drive home the big picture, even after reading both of the linked manuscripts-maybe my printer didn't indicate ATF6α?

This figure was corrupted during the upload process. We have included the correct version in the revised manuscript.

8) Shouldn't "its" in the title become "ATF6α"?

We agree —done.